# Low complexity domains of the nucleocapsid protein of SARS-CoV-2 form amyloid fibrils

Einav Tayeb-Fligelman[1,2,3,4], Jeannette T. Bowler [1,2,3,4], Christen E. Tai [1,2], Michael R. Sawaya [1,2,3,4,5], Yi Xiao Jiang [1,2,3,4], Gustavo Garcia Jr [6], Sarah L. Griner [1,2,3,4], Xinyi Cheng [1,2,3,4], Lukasz Salwinski [1,2,5], Liisa Lutter[1,2,3,4], Paul M. Seidler[1,2,11], Jiahui Lu[1,2,3,4], Gregory M. Rosenberg [1,2,3,4], Ke Hou [1,2,3,4], Romany Abskharon[1,2,3,4], Hope Pan [1,2,3,4], Chih-Te Zee[3], David R. Boyer [1,2,3,4], Yan Li [1,2], Daniel H. Anderson[1,2,3,4], Kevin A. Murray[1,2,3,4], Genesis Falcon[5], Duilio Cascio [5], Lorena Saelices [1,2,12], Robert Damoiseaux[6,7,8,9,10], Vaithilingaraja Arumugaswami[6,8,9], Feng Guo [1,2,10] & David S. Eisenberg [1,2,3,4,5,8] ✉

The self-assembly of the Nucleocapsid protein (NCAP) of SARS-CoV-2 is crucial for its function. Computational analysis of the amino acid sequence of NCAP reveals low-complexity domains (LCDs) akin to LCDs in other proteins known to self-assemble as phase separation droplets and amyloid fibrils. Previous reports have described NCAP's propensity to phase-separate. Here we show that the central LCD of NCAP is capable of both, phase separation and amyloid formation. Within this central LCD we identified three adhesive segments and determined the atomic structure of the fibrils formed by each. Those structures guided the design of G12, a peptide that interferes with the self-assembly of NCAP and demonstrates antiviral activity in SARS-CoV-2 infected cells. Our work, therefore, demonstrates the amyloid form of the central LCD of NCAP and suggests that amyloidogenic segments of NCAP could be targeted for drug development.

The Nucleocapsid protein (NCAP) of severe acute respiratory syndrome coronavirus 2 (SARS-CoV-2) is an RNA-binding protein that functions in viral replication by packaging the genomic viral RNA (vRNA) and aiding virion assembly[1–9]. During its function, NCAP engages in multivalent RNA–protein and protein–protein interactions and self-associates via several interfaces[10]. Increasing replication efficiency, NCAP forms concentrated protein–RNA compartments through a process of phase separation (PS)[1,2,4–8,10–12].

NCAP PS is enhanced in low salt buffers[4,5] and in the presence of zinc ions[2], and these PS droplets may exist in a liquid or solid-like state[1,2,4,8,11]. The liquid state of the droplets is favored by NCAP phosphorylation and is presumed to enable vRNA processing in the early stages of infection[4,8]. In contrast, non-phosphorylated NCAP oligomerizes and forms solid-like droplets, possibly to facilitate RNA packaging and nucleocapsid assembly in later stages[4,8].

[1]Department of Biological Chemistry, UCLA, Los Angeles, CA 90095, USA. [2]Molecular Biology Institute, UCLA, Los Angeles, CA 90095, USA. [3]Department of Chemistry and Biochemistry, UCLA, Los Angeles, CA 90095, USA. [4]Howard Hughes Medical Institute, Los Angeles, CA 90095, USA. [5]UCLA-DOE Institute of Genomics and Proteomics, UCLA, Los Angeles, CA 90095, USA. [6]Department of Molecular and Medical Pharmacology, UCLA, Los Angeles, CA 90095, USA. [7]Department of Bioengineering, UCLA, Los Angeles, CA 90095, USA. [8]California NanoSystems Institute, UCLA, Los Angeles, CA 90095, USA. [9]Eli and Edythe Broad Center of Regenerative Medicine and Stem Cell Research, UCLA, Los Angeles, CA 90095, USA. [10]Jonsson Comprehensive Cancer Center, UCLA, Los Angeles, CA 90095, USA. [11]Present address: Department of Pharmacology and Pharmaceutical Sciences, University of Southern California School of Pharmacy, Los Angeles, CA 90089-9121, USA. [12]Present address: Center for Alzheimer's and Neurodegenerative Diseases, Department of Biophysics, Peter O'Donnell Jr. Brain Institute, University of Texas Southwestern Medical Center, Dallas, TX 75390, USA. ✉e-mail: david@mbi.ucla.edu

The sequence of NCAP encompasses both RNA-binding and low-complexity domains. Low-complexity domains (LCDs) are protein segments comprised of a restricted subset of amino acid residues such as glycine, arginine, lysine, and serine[13–15]. Long mysterious in function, LCDs have recently been established to drive PS and form unbranched, twisted protein fibrils known as amyloid-like fibrils. Such behavior was observed in LCD-containing human RNA-binding proteins such as FUS, TDP-43, and hnRNPA2. By PS and amyloid formation, LCDs non-covalently link their parent proteins, and in some cases RNAs, into larger assemblies[13,16–18]. These larger assemblies are associated with the formation of subcellular bodies known variously as hydrogels[13,19], condensates[20], and membrane-less organelles[19]. In short, the self-association of several RNA-binding proteins has been shown to be driven at least in part by amyloid-like fibrils formed by their LCDs and to be a regulatory element of RNA metabolism in cells[19].

Motivating our study is a medical experience that even efficient vaccines rarely eradicate viral diseases and their legacies of morbidity and mortality[21], so COVID-19 therapies are needed. Along with others[10] we hold that NCAP of SARS-CoV-2 is a worthy drug target and that a better understanding of the structure and mechanism of action of NCAP may aid in drug development. NCAP is abundant in SARS-CoV-2-infected cells and its function is crucial for viral replication and assembly[10]. NCAP is also evolutionarily conserved in the coronavirus genus[10], which may render it as an effective target not only for COVID-19 treatments but possibly also for future coronavirus pandemics.

Here we show that NCAP possesses two fibril-forming LCDs, one central and one C-terminal. The central LCD forms Thioflavin-S (ThS)-positive PS droplets and amyloid fibrils that exhibit a characteristic diffraction pattern. At least three adhesive segments in this central LCD are capable of mediating amyloid typical interactions, and we elucidated the atomic structure of the fibrils formed by each. Guided by these structures, we designed a peptide that shifts NCAP to a less ordered mode of aggregation and investigated the peptide's effect on the infection of human cells by SARS-CoV-2.

## Results

### NCAP contains central and C-terminal LCDs

Using the SEG algorithm[22] we analyzed the sequence of NCAP and identified a 75-residue LCD (residues 175–249) within NCAP's central intrinsically disordered region, as well as a second, lysine-rich LCD of 19 residues (residues 361–379) within its C-terminal tail (CTT) (Fig. 1a, b). SEG is a widely used algorithm that identifies segments in a sliding window as either high or low complexity by statistically analyzing the amino acid distribution as a measure of sequence complexity[22]. While not all LCDs identified this way are capable of PS and amyloid formation, LCDs that do phase separate are readily identified by SEG[23,24]. In NCAP, those central and C-terminal LCDs, along with an N-terminal disordered region, flank the structured RNA-binding and dimerization domains of the protein (Fig. 1a).

### NCAP's LCDs participate in fibril formation

To assess possible amyloid formation of NCAP's LCDs and to identify adhesive segments that drive it, we expressed and purified NCAP and its LCD-containing segments in *E. coli*. Those segments included residues comprising NCAP's central LCD and surrounding residues (construct named LCD, residues 171–263) and a segment that includes the C-terminal LCD with the C-terminal tail and dimerization domain (construct DD-C$_{term}$, residues 257–419) (Fig. 1c). Only RNA-free protein fractions were combined at the last step of protein purification for use in subsequent experiments. We then verified that our purified full-length NCAP protein is capable of PS by mixing it with a 211-nucleotide 5′-genomic vRNA segment named hairpin-Site2 (S2hp; Supplementary Fig. 1, Supplementary Table 1) in the presence and absence of the PS enhancing $ZnCl_2$[2] (Supplementary Fig. 2a and Supplementary text). The S2 vRNA sequence was previously suggested to be a strong NCAP

cross-linking site[7], and we extended it by including the adjacent hairpin regions that improve binding to NCAP[25].

Using our recombinant protein system we found that NCAP's LCDs are capable of binding the amyloid-dye Thioflavin-T (ThT). In a ThT amyloid-formation kinetic assay performed over ~35 h of measurement (Fig. 1d, e), S2hp vRNA mixtures (in 4:1 protein: vRNA molar ratio) of the central LCD and the DD-C$_{term}$ segments of NCAP produced amyloid formation curves. Whereas the DD-C$_{term}$ + vRNA curve plateau after ~3 h of incubation, LCD + vRNA plateaus ~10 h after the start of measurements while producing a significantly higher fluorescence signal than that of DD-C$_{term}$. The full-length NCAP also exhibited increased ThT fluorescence over 5 h of measurement when mixed with S2hp vRNA, followed by a slight decrease in signal, possibly because of spontaneous disaggregation (Supplementary Fig. 2e). However, neither NCAP nor the DD-C$_{term}$ segment demonstrated a clear lag phase in their ThT curves. Also, in the absence of S2hp vRNA, we did not detect an increase in ThT fluorescence in any of the samples within 35 h of measurements. This suggests that vRNA promotes the formation of ThT-positive aggregates from those LCD-containing protein constructs, at least in the first 1.5 days of incubation.

Visualization of fibrils by electron microscopy (EM) confirmed the propensity of the LCD-containing constructs to adopt fibrillar morphologies (Fig. 1f and Supplementary Fig. 2c). To observe fibrils of NCAP and its LCD-containing segments by EM we increased protein concentration and incubated each protein separately for ~1–2 weeks with and without S2hp vRNA. Of note, under the conditions used for the kinetic ThT experiment (Fig. 1d, e and Supplementary Fig. 2e) we did not detect fibrils by EM, suggesting that the ThT experiment is more sensitive for the detection of amyloid-like aggregates or that ThT interacts with pre-fibrillar assemblies of the proteins. Other explanations, such as poor adherence of the protein fibrils to the EM grid and fibril reversibility are also reasonable. Nevertheless, with increased protein concentration and incubation time, fibrils were detected by EM both in the presence and absence of the vRNA. Fibrils of the DD-C$_{term}$ segment with vRNA are morphologically different than those grown in its absence, however, the central LCD segment produces amyloid-looking fibrils under both conditions. Indeed, concentrated LCD-only samples exhibit increased ThT fluorescence signal upon 6 and 11 days of incubation, but with large sample-to-sample variability (Fig. 1g). vRNA is, therefore, not essential for fibril formation and ThT binding, but may promote these processes.

The full-length NCAP also forms fibrillar morphologies in samples containing higher protein-to-vRNA ratio (40:1 protein:vRNA molar ratio), as well as when incubated with zinc ions in PBS (Supplementary Fig. 2b), and particularly in a low ionic strength buffer (Supplementary Fig. 2d) upon 3–6 days of incubation (as indicated in Supplementary Fig. 2). NCAP and also the DD-C$_{term}$ fibrils are much sparser in EM images compared to the central LCD, and their morphologies differ from those of the central LCD or canonical amyloid fibrils. Together, those observations suggest that NCAP and its LCD-containing segments, particularly the central LCD segment, are capable of forming aggregates of fibrillar morphologies as well as ThT-positive species.

### The central LCD forms amyloid typical fibrils

To examine the amyloid property of fibrils formed by NCAP and its LCD segment we used X-ray fiber diffraction. The X-ray fiber diffraction patterns of the central LCD showed a sharp reflection at 4.7 Å spacing and a diffuse reflection at 10 Å typical of amyloid fibrils. This is true for fibrils formed by the LCD alone (no RNA), and by LCD with S2hp vRNA or a non-specific RNA segment (Fig. 2a). This capacity of the central LCD segment of NCAP to stack into amyloid fibrils associates it with LCDs of other RNA-binding proteins that are involved in functional amyloid-formation and amyloid pathologies[13,16,18]. We were unable, however, to obtain a clear diffraction pattern from the full-length NCAP. This may be a result of low fibril concentration, as evident by EM

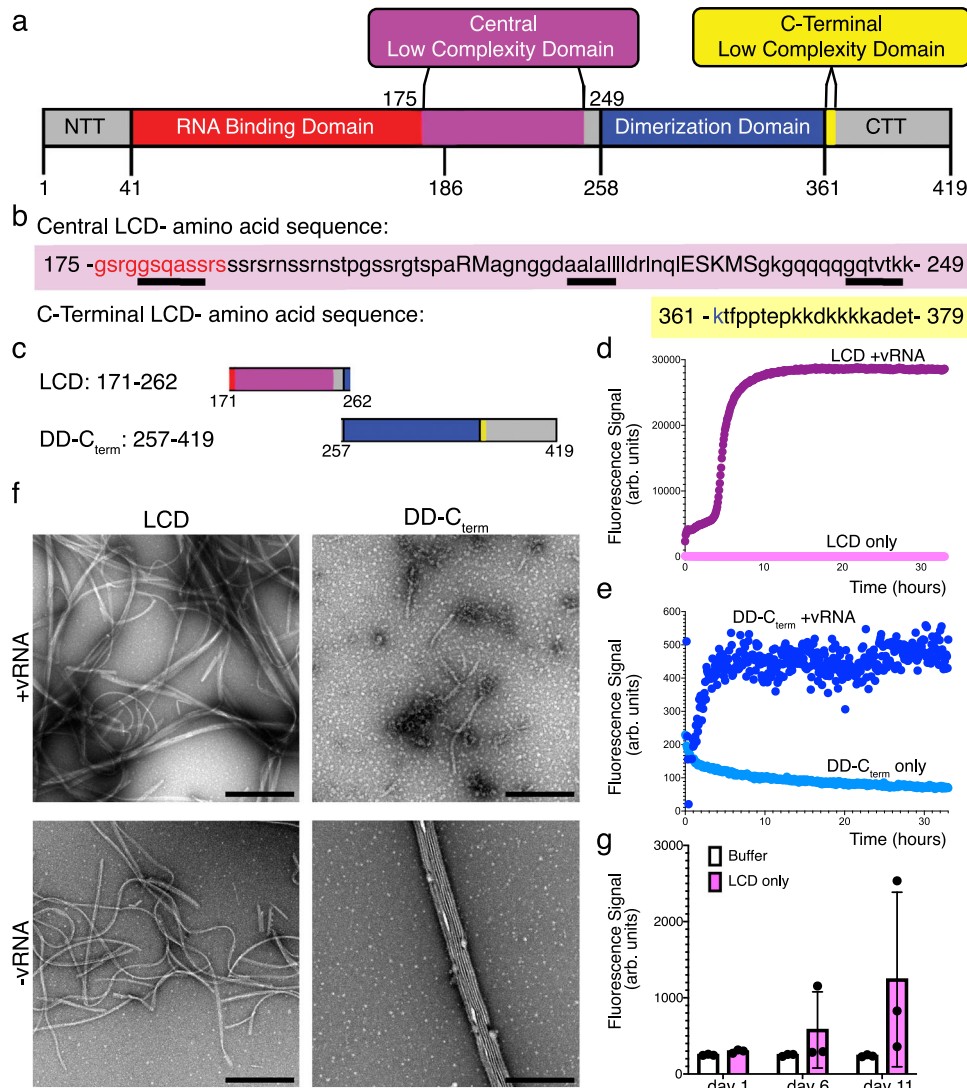

**Fig. 1 | NCAP's LCDs form fibrils and ThT-positive species. a** NCAP's domain organization. Domain definitions: N-terminal tail (NTT, gray), RNA-binding domain (red); Central low complexity domain (LCD, purple; residues 175–249), Dimerization domain (blue); C-terminal tail (CTT, gray). The C-terminal LCD is highlighted in yellow (residues 361–379). **b** Amino acid sequence of the central and C-terminal LCDs highlighted and colored according to the color scheme in (**a**). Lowercase letters represent residues of low complexity while capital letters represent non-low-complexity residues. No more than five interrupting non-low-complexity residues between strings of 10 or more low-complexity residues were allowed. Steric-zipper forming sequences that are discussed below are underlined in the sequence of the central LCD. **c** Protein segments used in this study are abbreviated as LCD, consisting of the central LCD and surrounding residues, and as DD-C$_{term}$, consisting of

the dimerization domain (DD) and the C-terminal tail, including the C-terminal LCD. The LCD and DD-C$_{term}$ segments are colored according to the color scheme in (**a**). **d** and **e** ThT fibril formation kinetic assays of the LCD (**d**) and DD-C$_{term}$ (**e**) segments incubated with (purple/navy, respectively) and without (pink/light blue, respectively) hairpin-Site2 (S2hp) viral RNA (vRNA). **f** Fibril formation from concentrated LCD and DD-C$_{term}$ samples observed by negative stain EM after 6 days of incubation with and without S2hp vRNA. Scale bar = 500 nm. **g** Endpoint ThT fluorescence measurements of concentrated LCD-only samples (pink) and buffer-only controls (white) at days 1, 6, and 11 of incubation. Dots indicate individual data points and bars represent mean values ± SD. $n = 3$ samples. Source data for panels **d**, **e**, and **g** are provided as a Source Data file.

(Supplementary Fig. 2c), and/or from fibril decomposition during washing steps meant to eliminate salts from the sample.

The central LCD segment of NCAP also readily forms unbranched fibrils in the presence of short, unstructured vRNA types such as the Site1 (S1), Site1.5 (S1.5) and S2 segments, as well as with a non-specific RNA segment of a similar length (Fig. 2b: Supplementary Table 1), and even with no RNA (Fig. 1f). When the LCD segment is incubated for one day with either S1 or S2 vRNA segments, the LCD produces heavily-stained clusters with fibrils protruding from their edges, but these clusters disperse after 4 days of incubation. Such behavior is not observed with S1.5 vRNA or the non-specific RNA segment (Fig. 2b). This may suggest that the LCD fibril growth process may be altered by the RNA sequence. Overall, the amyloid formation of the central LCD

offers that this region could potentially promote ordered self-assembly of NCAP under the appropriate conditions.

## NCAP's central LCD forms PS droplets and solid particles

Next, we examined the capacity of the central LCD to form PS droplets with different S2hp vRNA concentrations and followed the behavior and character of the droplets over time in the presence of the amyloid dye Thioflavin-S (ThS) using light and fluorescence microscopy. In samples of 4:1 and 40:1 LCD: S2hp vRNA molar ratios we visualized PS droplets that gradually transition into rough, less circular, seemingly solid particles (Fig. 3a, b). In the 40:1 LCD: S2hp sample, PS droplets form and begin to fuse within 30 min of incubation, and ThS partitions into the droplets and produces rather bright fluorescence (Fig. 3a, c).

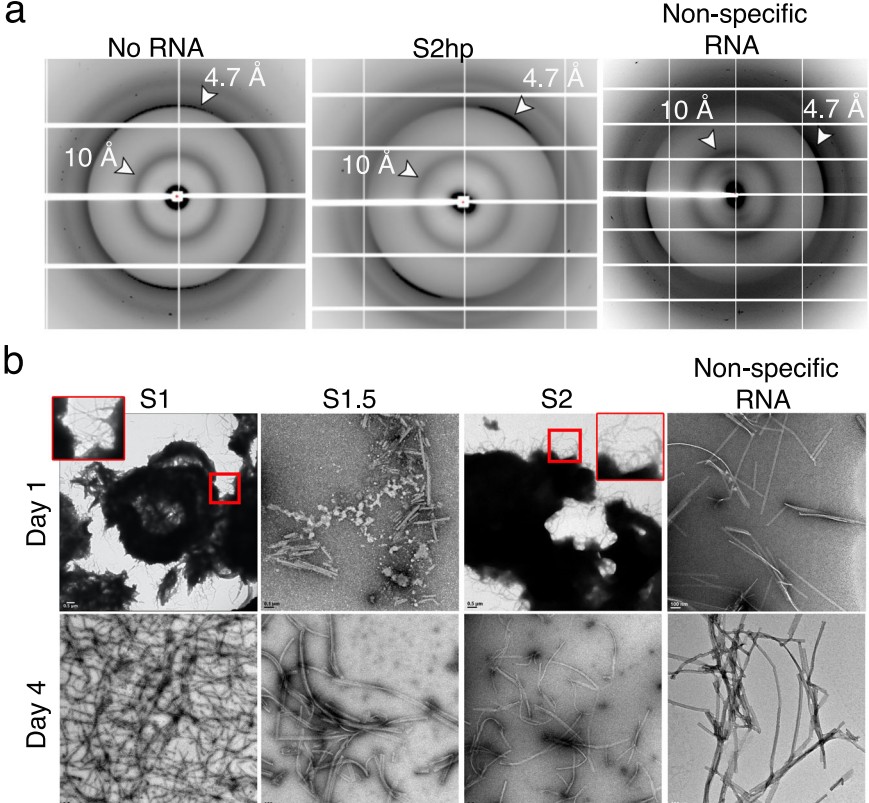

**Fig. 2 | The central LCD segment of NCAP demonstrates amyloid-like characteristics. a** X-ray diffractions of LCD-only fibrils (No RNA), and LCD fibrils grown with hairpin-Site2 (S2hp) vRNA or non-specific RNA (antisense siDGCR8-1), show amyloid-characteristic 4.7 and 10 Å reflections labeled by white arrows. **b** Negative stain EM micrographs of LCD fibrils grown in the presence of the short vRNA segments Site1 (S1; 11 nucleotides), Site 1.5 (S1.5; 22 nucleotides) and Site2 (S2; 22 nucleotides), as well as with a non-specific RNA (antisense siDGCR8-1). All RNA sequences are given in Supplementary Table 1. This figure shows that the central LCD produces amyloid-typical fibrils in the absence and presence of different RNA segments and that the RNA sequence may influence the morphology of the LCD assemblies over time.

Upon 2 h of incubation, larger asymmetric droplets appear, and after 6 h, filamentous structures decorate the droplets. Within 4 days of incubation, the droplets transform into what appear as solid-like filamentous particles. At a higher S2hp concentration (4:1 LCD: S2hp molar ratio), small PS droplets appear after ~30 min, but those droplets show almost no ThS fluorescence (Fig. 3a, c). Additional PS droplets form after 2 h of incubation and a weak ThS signal is detected. However, after 6 h incubation, and even more predominantly after 4 days, most droplets convert into brightly fluorescent particles (Fig. 3a, c).

An analysis of LCD assemblies (droplets and solid-like particles) from a series of light microscope images taken at different time points of incubation shows that the mean area of the 40:1 LCD:vRNA assemblies somewhat increases upon the transition from liquid droplets to the fibrous looking particles. The median value of the mean circularity of the assemblies (weighted by the size of the droplet/particle) drops by ~60% between the first (day 1) and last (day 4) measurements (Fig. 3b, left). A similar analysis of the 4:1 LCD: vRNA sample revealed a greater increase in the mean area of the assemblies upon 4 days of incubation, and a greater decrease of ~80% in the median value of the mean circularity (Fig. 3b, right), suggesting a massive transition of circular liquid droplet into large, amorphous, solid-like particles. Quantification of the mean ThS fluorescence from images taken at 0.5 h and 4 days of incubation of both samples show a ~4-fold increase in ThS fluorescence in the 40:1 LCD:S2hp sample, and ~58-fold increase in fluorescence intensity in the 4:1 LCD:S2hp ratio (Fig. 3c). Here too, no fibrils could be detected by EM at the concentration and incubation times used for the PS assay.

In a separate experiment, we also followed the aggregation of the central LCD segment when incubated alone or with S2hp vRNA (in 4:1 respective ratio) by measuring turbidity (Fig. 3d). We detected elevated turbidity of the LCD + vRNA sample at the beginning of the measurement, as opposed to the LCD only sample that was not turbid. This offers that the central LCD immediately aggregates upon mixing with vRNA. The LCD + vRNA sample shows biphasic behavior, with a decrease in turbidity between 0 and 5.5 h of incubation, followed by a renewed increase. This biphasic behavior of the 4:1 LCD:S2hp vRNA sample may be related to the transition from liquid droplets to solid particles visualized in this sample between 2 and 6 h of incubation (Fig. 3a). Overall, our results indicate that the central LCD of NCAP forms ThS-positive PS droplets that transition from circular liquid droplets to fibrous or amorphous solid-like particles, and that the RNA concentration governs the kinetics of this process and the morphology of the assemblies.

**Structures of LCD-derived steric-zipper-forming segments**

To interfere with the self-assembly of the LCD segment, and thereby possibly of NCAP, we seek structural information of specific amyloid-like LCD sequences. Amyloid fibrils are stabilized by pairs of tightly mating β-sheets, with zipper-like interfaces termed steric zippers that can be predicted by a computer algorithm[26] [https://services.mbi.ucla.edu/zipperdb/]. Within the central LCD, we identified (Supplementary Fig. 3a) and crystallized three such steric zipper-forming segments: $_{179}$GSQASS$_{184}$, $_{217}$AALALL$_{222}$, and $_{243}$GQTVTK$_{248}$. X-ray structures confirmed that each segment forms amyloid-like fibrils composed of pairs of β-sheets stabilized by steric zipper interfaces (Fig. 4, and

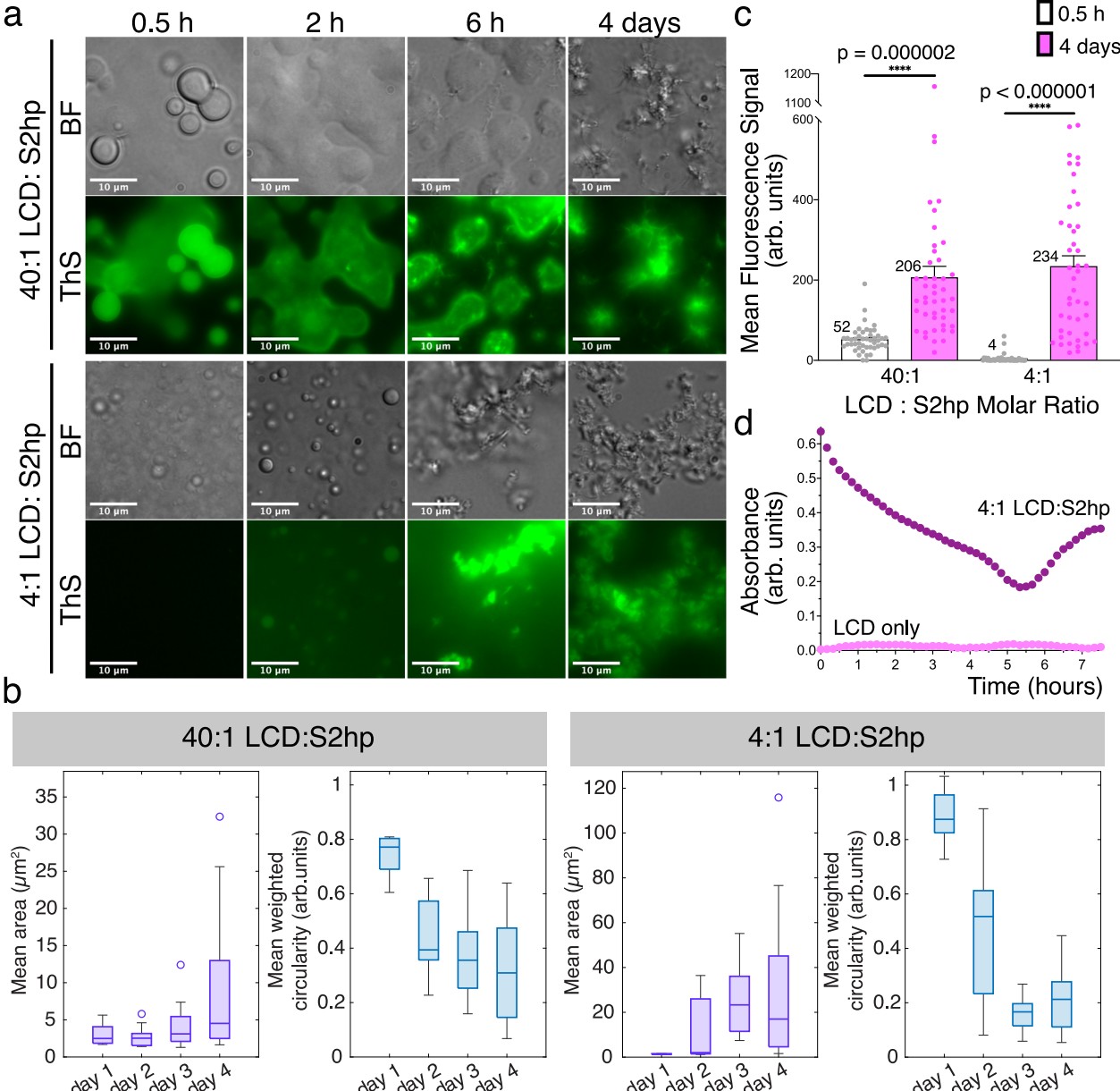

**Fig. 3 | The central LCD segment of NCAP forms ThS fluorescent PS droplet that transition into amorphous and fibrous solid-like particles. a** Brightfield (BF) and Thioflavin-S (ThS) fluorescence (green) microscopy images of 40:1 and 4:1 LCD: hairpin-Site2 (S2hp) vRNA molar ratio mixtures incubated for ~0.5, 2, 6 h and 4 days. **b** Mean area (purple) and mean circularity (blue; normalized to particle size) of droplets and particles quantified from a series of light microscopy images of 40:1 and 4:1 LCD: S2hp mixtures imaged at day 1–4 of incubation. The experiment was performed in three biological repeats, each with technical triplicates. Five images were collected for every technical replicate. Boxplots show the 25th percentile, median, and 75th percentile of the mean values for triplicate experiments. The whiskers extend to the most extreme data points. Observations beyond the whisker length, shown as circles, are values more than 1.5 times the interquartile range beyond the bottom or top of the box (n = 9 replicates). **c** Mean ThS fluorescence signal measured from background-subtracted fluorescence microscopy images taken from 40:1 and 4:1 LCD: S2hp mixtures at 0.5 h (white) and 4 days (pink) of incubation. The experiment was performed in three biological repeats, each with technical triplicates. Five images were collected for every technical replicate. Data from all repeats were combined for the quantification. The dots are of individual data points and the bars represent mean values ± SEM (n = 45 images). Statistical significance was calculated in Prism using an unpaired two-tailed t-test with Welch's correction. The p values are indicated with numbers and stars—****p < 0.0001. Welch's corrected t = 5.377/ 8.597 and df = 46.59/44.33 for 40:1 and 4:1 LCD: S2hp samples, respectively. **d** Time-dependent shift in turbidity of LCD only (pink) and 4:1 LCD: S2hp (purple) solutions evaluated by measuring absorbance at 600 nm. Source data for panels **b–d** are provided as a Source Data file.

Supplementary Figs. 4 and 5; Table 1). GSQASS and GQTVTK segments both form parallel, in-register β-sheets, whereas the AALALL segment is crystalized in two forms, both with antiparallel β-sheets[27]. The weaker zipper interface of the second form incorporates polyethylene glycol (Supplementary Fig. 4), and we do not consider it further.

Solvation-free energy calculations based on our crystal structures (Supplementary Table 2) suggest that the AALALL steric-zipper is the most stable of the three, consistent with its predominance of hydrophobic residues. GSQASS and GQTVTK, on the contrary, contain mostly polar residues (Fig. 4c). The AALALL segment also overlaps with a region predicted to participate in context-dependent interactions of NCAP (Supplementary Fig. 3b, residues 216–221), namely interactions that change between disordered and ordered modes as a function of cellular environment and protein interactors and are likely to be

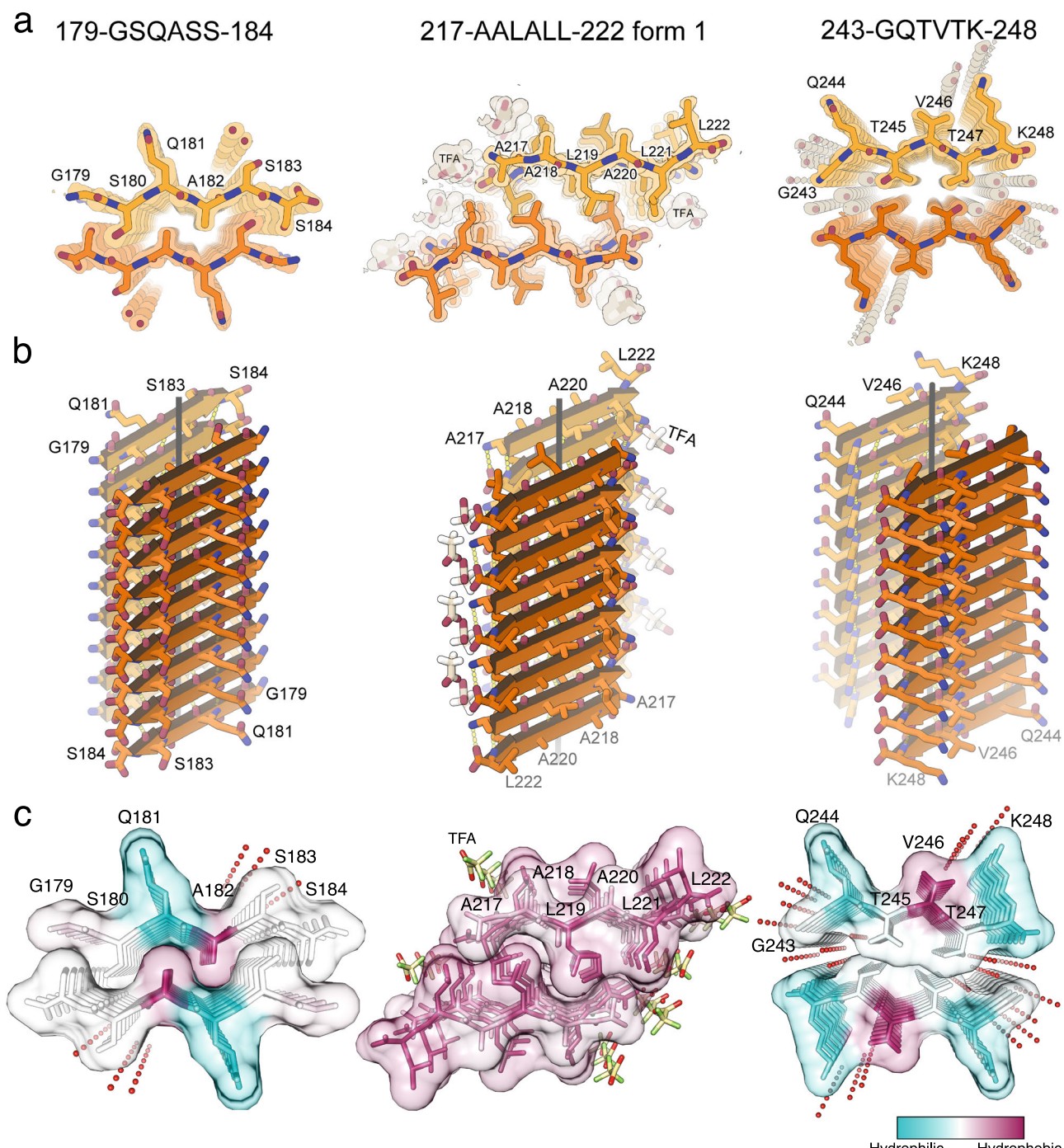

**Fig. 4 | Atomic structures of amyloid-like association of NCAP segments revealed by crystallography. a** Quality of the fit of each atomic model to its corresponding simulated annealing composite omit maps[92]. The maps are contoured at the 1.0 sigma level. All structural features are well defined by the density. The view is down the fibril axis. Each chain shown here corresponds to one strand in a β-sheet. Thousands of identical strands stack above and below the plane of the page making ~100 micron-long β-sheets. The face of each β-sheet of AALALL (PDB 7LTU) [https://doi.org/10.2210/pdb7LTU/pdb] (form 1) is symmetric with its back. However, GSQASS (PDB 7LV2) [https://doi.org/10.2210/pdb7LV2/pdb] and GQTVTK (PDB 7LUZ) [https://doi.org/10.2210/pdb7LUZ/pdb] each reveal two distinct sheet–sheet interfaces: face-to-face and back-to-back. The tighter associated pair of sheets is shown in this figure. **b** 18 strands from each of the steric zippers at a view nearly perpendicular to the fibril axis. GSQASS and GQTVTK are parallel, in-register sheets, mated with Class 1 zipper symmetry. The AALALL zippers are antiparallel, in register sheets, mated with Class 7 zipper symmetry. Trifluoroacetic acid (TFA) appears in the AALALL-form 1 steric zipper, and polyethylene glycol (PEG) binds form 2 (PDB 7LUX [https://doi.org/10.2210/pdb7LUX/pdb] (form 2); Supplementary Fig. 4). Carbon atoms in **a** and **b** are shown in orange and heteroatoms are colored by atom type. Water molecules are shown as red dots. **c** Steric zipper structures (same order as in **a**) viewed down the fibril axis with residues colored according to the Kyte and Doolittle hydrophobicity scale (bottom right) shown with semi-transparent surface representation to emphasize the shape complementarity and tight fit between the β-sheets. Images in **c** were rendered with UCSF Chimera[90]. A stereo view of all structures is given in Supplementary Fig. 5.

**Table 1 | Crystallographic data collection and refinement statistics from SARS-CoV-2 NCAP segments**

| Segment | [179]GSQASS[184] | [217]AALALL[222] Form 1 | [217]AALALL[222] Form 2 | [243]GQTVTK[248] |
|---|---|---|---|---|
| *Data collection* | | | | |
| Beamline | APS 24-ID-E | APS 24-ID-E | APS 24-ID-E | APS 24-ID-E |
| Space group | $P2_12_12_1$ | P1 | $P2_12_12$ | $P2_1$ |
| Resolution (Å) | 1.30 (1.39–1.30)[a] | 1.12 (1.18–1.12) | 1.30 (1.36–1.30) | 1.10 (1.17–1.10) |
| Unit cell dimensions: $a,b,c$ (Å) | 4.77, 13.60, 42.44 | 9.45, 11.34, 20.27 | 44.46, 9.54, 10.95 | 19.57, 4.78, 22.03 |
| Unit cell angles: $\alpha,\beta,\gamma$ (°) | 90.0, 90.0, 90.0 | 74.9, 79.1, 67.8 | 90.0, 90.0, 90.0 | 90.0, 94.0, 90.0 |
| Measured reflections | 1833 (338) | 5371 (323) | 4666 (550) | 4677 (344) |
| Unique reflections | 809 (139) | 2270 (136) | 1234 (139) | 1726 (170) |
| Overall completeness (%) | 93.2 (95.9) | 78.4 (31.1) | 93.0 (84.8) | 87.1 (50.9) |
| Overall redundancy | 2.3 (2.4) | 2.4 (2.4) | 3.8 (4.0) | 2.7 (2.0) |
| Overall $R_{merge}$ | 0.126 (1.04) | 0.084 (0.397) | 0.105 (0.808) | 0.085 (0.446) |
| $CC_{1/2}$ | 99.7 (56.7) | 98.5 (89.2) | 99.7 (54.4) | 99.5 (84.3) |
| Overall I/δ | 3.5 (0.7) | 5.9 (2.0) | 5.9 (1.8) | 6.0 (1.4) |
| Refinement | | | | |
| $R_{work}/R_{free}$ | 0.259/0.253 | 0.158/0.197 | 0.217/0.248 | 0.133/0.177 |
| RMSD bond length (Å) | 0.015 | 0.009 | 0.010 | 0.009 |
| RMSD angle (°) | 1.4 | 1.3 | 1.6 | 1.5 |
| Number of segment atoms | 40 | 180[b] | 40 | 93[b] |
| Number of water atoms | 2 | 1 | 1 | 12 |
| Number of other solvent atoms | 0 | 21 | 14 | 0 |
| Average $B$-factor of peptide (Å²) | 12.3 | 12.3 | 14.2 | 8.2 |
| Average B-factor of water (Å²) | 19.9 | 12.8 | 26.6 | 24.7 |
| Average $B$-factor other solvent (Å²) | N/A | 20.8 | 27.3 | N/A |
| PDB ID code | 7LV2 [https://doi.org/10.2210/pdb7LV2/pdb] | 7LTU [https://doi.org/10.2210/pdb7LTU/pdb] (form 1) | 7LUX [https://doi.org/10.2210/pdb7LUX/pdb] (form 2) | 7LUZ [https://doi.org/10.2210/pdb7LUZ/pdb] |

[a]Numbers in parentheses report statistics in the highest resolution shell.
[b]Count includes hydrogen atoms.

responsible for the formation of amyloid fibrils within liquid droplets[28]. For drug design, we pursued AALALL and GQTVTK as targets but excluded GSQASS because it resembles LCDs found in the human proteome[29].

### A structure-based disruptor of NCAP's PS exhibits antiviral activity

To modulate NCAP's self-assembly we exploited the propensity of NCAP's LCD to form steric-zipper structures. Guided by our amyloid-spine structures we screened an array of peptides, each designed to interact with a specific steric-zipper forming segment. We have found such peptides to inhibit the aggregation and prion-like seeding of other amyloid-forming proteins (e.g. refs. [30–33]). To design the steric-zipper targeting disruptors of NCAP self-assembly we implemented two approaches: sequence/structure-based design and Rosetta-based modeling[34]. Both approaches produce sequences that bind strongly to our steric zipper structure targets and contain bulky residues that block the interactions of additional NCAP molecules via this interface (Fig. 5a).

Screening of a panel of our designed peptides in vitro revealed that a peptide we named G12 disrupts NCAP's PS. G12 is a D-amino acid peptide with the sequence d-(rrffmvlm), designed against the AALALL steric zipper-forming segment (Fig. 5a; Supplementary Table 3). Increasing concentrations of G12 disrupt the formation of circular NCAP PS droplets and instead promote the formation of large network-like aggregates as judged by light microscopy (Fig. 5b, c and Supplementary Fig. 6).

We then proceeded to test G12's antiviral activity in HEK293 cells that express the human ACE2 receptor (HEK293-ACE2 cells). First, we verified that HEK293-ACE2 cells transfected with FITC-labeled G12 show that G12 remains soluble and diffuse in the cytoplasm for at least 24 h (Supplementary Fig. 7). Next, we used quantitative immunofluorescence labeling to detect the percentage of SARS-CoV-2 infection in cells transfected with increasing concentrations of G12 or a vehicle only control. The percentage of infected cells in each G12-treated culture was normalized to the infected vehicle-only control (Fig. 5d and Supplementary Fig. 8). Cytotoxicity was tested with the same cells and G12 concentrations using the LDH toxicity assay (Fig. 5d, red curve). Whereas G12 concentrations lower than 6 µM slightly increase the relative percent infectivity of treated cells, in the range of ~6–16 µM, G12 exhibits dose-dependent antiviral activity while reducing the amount of virus detected in the culture by up to ~50% without inflicting cytotoxicity (Fig. 5d and Supplementary Fig. 8). Since G12 is dissolved in DMSO we could not test higher G12 concentrations in this cell-based assay to obtain a complete dose–response curve, however fitting a non-linear regression model to our data allowed a rough $IC_{50}$ estimation of 7–11 µM (Fig. 5d). We, therefore, suggest that G12 serves as a proof-of-concept showing that by targeting amyloidogenic segments within the central LCD of NCAP we interfere with NCAP's self-assembly and thereby the viral life cycle.

### Discussion

The NCAP protein of SARS-CoV-2 belongs to the subclass of fibril-forming proteins that contains both an RNA-binding domain and LCDs

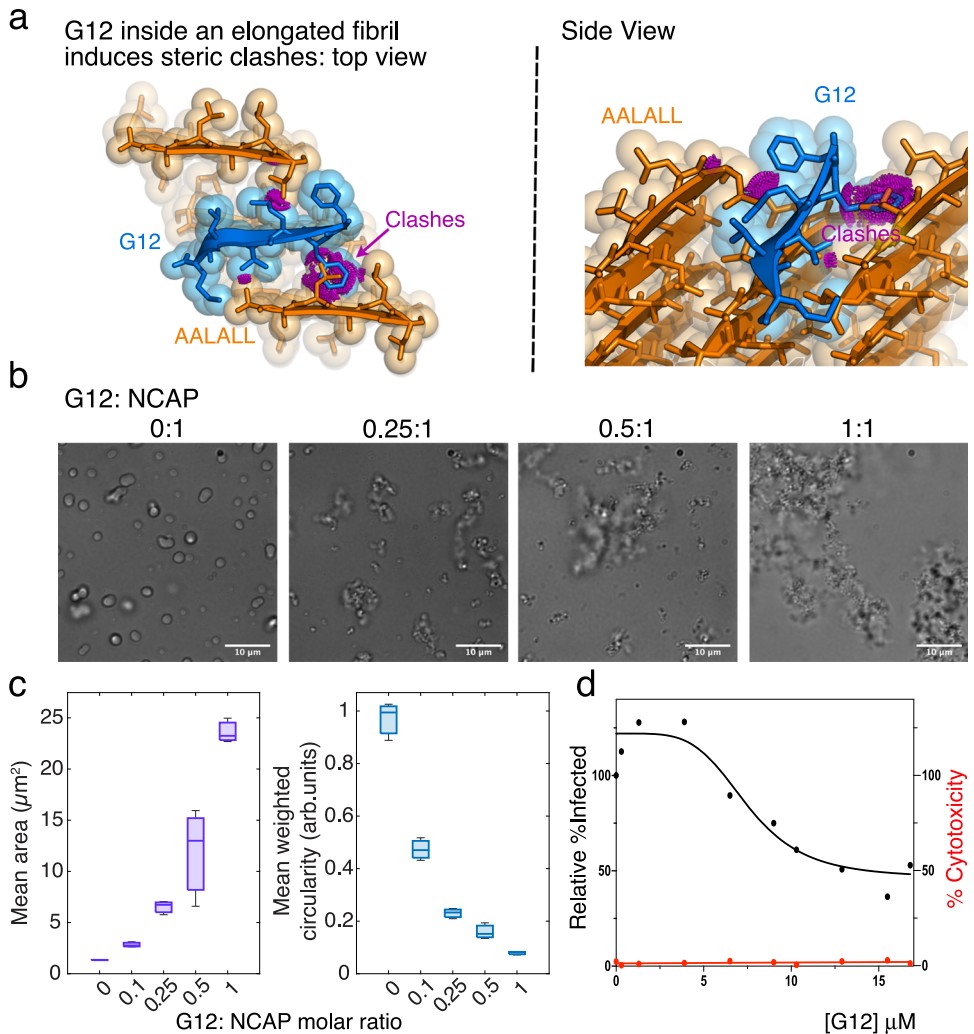

**Fig. 5 | Design and evaluation of NCAP's self-assembly disruptor, G12. a** The Rosetta-based design of G12 templated by the AALALL X-ray crystal structure form 1 (Fig. 4; Table 1). Model of the G12 (blue) capping an AALALL fibril (orange). The top view is down the fibril axis and the side view is tilted from the axis. Additional AALALL strands are shown adjacent to the bound G12 to illustrate their steric clashes (magenta). **b** Differential interference contrast (DIC) images of NCAP + S2hp mixtures incubated in the absence (0:1) and presence of increasing concentrations of G12 revealing the PS disrupting activity of G12. **c** Mean area (purple) and mean circularity (blue; normalized to particle size) of droplets and particles quantified from a series of light microscopy images of NCAP + S2hp mixtures with increasing concentrations of G12. The experiment was performed in three biological repeats, each with technical triplicates. Five images were collected for every technical replicate. A representative plot is presented. In boxplots the central mark indicates the median, and the bottom and top edges of the box indicate the 25th and 75th percentiles, respectively. The whiskers extend to the most extreme data points ($n = 3$ replicates). **d** Dose–response analysis of HEK293-ACE2 cells treated with 10 different concentrations of G12 and fitted with a nonlinear regression model (black line). The 95% confidence interval of the IC50 for G12 was estimated to be between 7 and 11 μM. Cytotoxicity testing of G12 at each concentration (red line) indicates that G12 is non-toxic. Each dot represents the mean value of three technical replicates. Source data for panels **c** and **d** are provided as a Source Data file.

(Fig. 1a, b). NCAP undergoes PS[1–8] (Supplementary Fig. 2a), and as we show here, its central LCD forms amyloid-like fibrils (Figs. 1 and 2).

The central LCD of NCAP forms fibrils with the long and structured S2hp vRNA segment (Fig. 1 and Supplementary Fig. 1; Supplementary Table 1), with various short, single-stranded RNA sequences (Fig. 2; Supplementary Table 1), and also with no RNA (Fig. 1f). This suggests that specific LCD–RNA interactions are not required for LCD-amyloid formation. Nevertheless, the LCD does bind to at least S2hp vRNA[25], and LCD fibril maturation is influenced by the RNA sequence and length (Fig. 2b), so LCD–RNA interactions play a role. The LCD segment is highly positively charged (Fig. 1b), especially in its non-phosphorylated form. Therefore we expect it to engage in non-specific polar interactions with the negatively charged RNA, which in turn may promote the accumulation of LCD molecules, including through PS formation (Fig. 3), and their amyloid-like assembly (Fig. 2a).

The amyloid-like characteristics of the central LCD of NCAP are similar to those of the LCDs of FUS[35,36], hnRNPA2[37], TDP-43[38], and other RNA-binding proteins that are involved in RNA metabolism in eukaryotic cells[13,17], and under certain circumstances, also in amyloid-associated pathologies[19]. This equivalent ability of the LCD of NCAP to PS and stack into amyloid-like structures in the presence of RNA proposes its potential function in the yet elusive mechanism of NCAP self-assembly.

Full-length NCAP is capable of only sparse fibril formation in the presence and absence of S2hp vRNA and with ZnCl$_2$ (Supplementary Fig. 2). Whereas fibrils formed in the presence of S2hp do not exhibit amyloid-typical morphology (Supplementary Fig. 2b, c), the NCAP + S2hp aggregates produce a ThT amyloid formation curve, but it lacks a lag phase (Supplementary Fig. 2e). Short or absent lag phase in ThT curves may result from the existence of pre-formed amyloid seeds in the tested sample[39], or from a fast pickup of the ThT signal prior to

starting the measurements. The latter may be reasonable given that NCAP rapidly aggregates and becomes turbid in the presence of S2hp (Supplementary Fig. 2f). In a parallel study, we show that the structured regions of S2hp are essential for strong binding to NCAP, whereas S2 and other short, single-stranded RNA segments bind to it weakly[25]. Here, we detected fibrils of NCAP with S2hp (Supplementary Fig. 2b, c), but its LCD segment is also capable of forming fibrils in the presence of the short, unstructured vRNA segments S1, S1.5, and S2 (Fig. 2b; Supplementary Table 1). We, therefore, speculate that robust amyloid formation of full-length NCAP requires strong interactions with specific vRNA sequences and/or co-factors that we are yet to identify.

The amyloid formation of the central LCD of NCAP is attributed to at least three adhesive peptide sequences: $_{179}$GSQASS$_{184}$, $_{217}$AALALL$_{222}$, and $_{243}$GQTVTK$_{248}$ (Fig. 4 and Supplementary Fig. 3a). $_{179}$GSQASS$_{184}$ and $_{243}$GQTVTK$_{248}$, are predominantly polar (Fig. 4c), similar to the highly polar reversible amyloid fibrils formed by the LCDs of FUS and hnRNPA2[40]. The segment $_{179}$GSQASS$_{184}$ is part of a conserved serine/arginine (SR)-rich region (residues 176–206)[4] and it includes the two phosphorylation sites S180 and S184[11]. Phosphorylation of the SR-rich region facilitates the transformation of NCAP's PS droplets from a solid to a liquid-like state during viral genome processing. The non-phosphorylated protein, however, is associated with solid PS droplets and nucleocapsid assembly[8]. Both S180 and S184 face the dry, tight interface formed between the β-sheets in the structure of $_{179}$GSQASS$_{184}$ (Fig. 4a). Phosphorylation of those residues is indeed likely to reverse the solid, amyloid-like packing of this segment. Of note, all results in this paper showing the ordered, solid-like mode of aggregation were obtained with non-phosphorylated proteins and peptides.

The second adhesive segment, $_{217}$AALALL$_{222}$, is highly hydrophobic and produces the most stable steric-zipper structure (Supplementary Table 2). $_{217}$AALALL$_{222}$ is also predicted to help switch between disordered and ordered modes of protein aggregation as a factor of cellular environment and protein interactors (Supplementary Fig. 3c, residues 216–221)[28]. Those properties of $_{217}$AALALL$_{222}$ render it an important target for the disruption of NCAP's self-assembly. The $_{243}$GQTVTK$_{248}$ segment, however, resembles sequences in LCDs found in the human proteome[29], and is therefore a poor target for drug design.

The self-assembly of NCAP is crucial for RNA packaging and SARS-CoV-2 replication[10]. The amyloid formation of NCAP's LCD is a form of NCAP self-assembly, but it is yet unclear whether NCAP forms and functions as amyloid in the viral life cycle. Nevertheless, PS-mediated self-assembly of NCAP was shown to occur in NCAP-transfected and SARS-CoV-2 infected cells[4,7,11,41]. By targeting the amyloidogenic segment $_{217}$AALALL$_{222}$ (Fig. 4 and Supplementary Figs. 4 and 5) with G12, we inhibited the PS formation of NCAP in vitro (Fig. 5b, c). G12 is a peptide designed to interact and block the $_{217}$AALALL$_{222}$ interface by exploiting the tendency of this segment to form steric-zipper structures (Fig. 5a). G12 is, however, incapable of complete disruption of NCAP self-assembly, perhaps because assembly is guided by several proteins interfaces[10]. Evaluation of G12 in SARS-CoV-2-infected cells revealed dose-dependent antiviral activity in concentrations of 6–16 μM without inflicting cytotoxicity (Fig. 5d). G12 concentrations lower than 6 μM, however, led to increased viral infection in treated cells. We speculate that when administered in subeffective concentrations, G12 partitions into NCAP droplets and increases NCAP's effective concentration which possibly promotes self-assembly and formation of new virions. When administered in proper concentrations, we anticipate that the antiviral activity of G12 results from its interference with the self-assembly of NCAP, as designed, leading to poor RNA packaging and viral particle assembly.

The three steric-zipper-forming segments we identified in this work are conserved between the NCAPs of SARS-CoV-2 and SARS-CoV.

The only exception is alanine in position 217 in the sequence of SARS-CoV-2 which is replaced by threonine in the NCAP of SARS-CoV (Supplementary Fig. 9a). A ZipperDB[26] [https://services.mbi.ucla.edu/zipperdb/] calculation on the LCD of the NCAP of SARS-CoV revealed that this threonine shifts the steric-zipper forming segment to the hydrophobic ALALLL sequence (with Rosetta free energy score of −24.700) that is aligned and conserved with residues 218–223 in the NCAP of SARS-CoV-2. This suggests that the LCD in the NCAP of SARS-CoV may also form amyloids, and that future SARS coronaviruses might share this targetable property. A SEG analysis[22] performed on the sequence of the NCAPs of a number of α, β and γ coronaviruses from various species showed that many of these viruses contain LCDs that could potentially participate in amyloid formation (Supplementary Fig. 9b). This suggests that amyloid formation of NCAP LCDs is a general mechanism of action and a common targetable trait in coronaviruses.

Despite the high conservation of NCAP[10], some mutations have been identified in strains that emerged since the initial SARS-CoV-2 outbreak in Wuhan, China. To date, no NCAP mutations were detected within our amyloid steric-zipper spine segments: $_{179}$GSQASS$_{184}$, $_{217}$AALALL$_{222}$, and $_{243}$GQTVTK$_{248}$. Nevertheless, some mutations were detected within the central LCD, including the prevalent R203K/M, G204R/M, and T205I substitutions[42–44]. The R203K/G204R mutants exhibit higher PS propensity compared to the Wuhan variant[41], and the R204M mutation promotes RNA packaging and viral replication in the delta variant[45]. Also interesting are the G214C (Lambda variant) and G215C (Delta variant) substututions[42–44] that are adjacent to the $_{217}$AALALL$_{222}$ steric-zipper segment. The Delta variant spread faster and caused more infection compared to its predecessors[46–49]. The Delta variant also carries a D377Y mutation in the C-terminal LCD segment of NCAP. It is possible that mutations in NCAP's LCD enhance amyloid formation, similarly to mutations in other RNA-binding proteins[35,50–52]. This is important to explore since amyloid fibrils are associated with numerous dementias and movement disorders[53,54]. Amyloid cross-talk and hetero-amyloid aggregation, including between microbial and human amyloid proteins (e.g. refs. 55–58), is a well-known phenomenon that is postulated to exacerbate amyloid pathology[59].

The possible connection of amyloid formation of NCAP to neurodegeneration was already recently suggested. NCAP was shown to interact and accelerate the amyloid formation of the Parkinson's disease-related protein, α-synuclein, which may explain the correlation between Parkinsonism and SARS-CoV-2 infection[60]. NCAP was also shown to partition into PS droplets[5] and accelerate amyloid formation[61] of FUS, TDP-43, hnRNPA1, and hnRNPA2. In certain forms, those proteins are associated with neurodegenerative and movement disorders[19]. In SARS-CoV-2 infected cells, NCAP impairs the disassembly of stress granules into which it partitions, and in cells expressing an ALS-associated mutant of FUS, NCAP enhances FUS aggregation into amyloid-containing puncta[61]. Those observations, together with the capacity of NCAP's central LCD to form amyloid, call for further investigation of the possible NCAP-amyloid formation and regulation in SARS-CoV-2-infected cells, and of the possible involvement of NCAP in amyloid cross-talk and human neurodegeneration.

Our study of the amyloid formation of NCAP expands an emerging class of known amyloid-forming viral proteins. In the Influenza A virus, the full-length and N-terminal segment of the PB1-F2 protein form cytotoxic amyloid fibrils when mixed with liposomes, and the C-terminal segment forms cytotoxic amyloid oligomers[62]. A 111-residue segment from the V protein of Hendra virus, a respiratory virus that may progress in humans to severe encephalitis, was shown to undergo a liquid-to-hydrogel transition of its PS droplets and to produce amyloid-like fibrils[63]. The RIP-homotypic interaction motif containing segments of the herpes simplex virus 1 (HSV-1) protein ICP6[64], the

murine cytomegalovirus protein M45[65] and the varicella-zoster virus protein ORF20[66] are capable of forming heteromeric amyloid complexes with host proteins. Other examples of amyloid-forming peptide segments include avibirnavirus viral protease that contributes to protease self-assembly[67], peptides from the fiber protein of adenovirus[68,69], and a nine-residue peptide from the C-terminus of the SARS-CoV envelope protein[70]. Recent studies also showed the amyloidogenic properties of various segments of the spike protein[71], and other regions in the proteome[72] of SARS-CoV-2. None of these previously studied viral amyloids, however, was associated with NCAPs. Nevertheless, LCDs and prion-like sequences, such as those that exist in NCAP[5] were identified in over two million eukaryotic viruses[73]. Therefore, our finding of the amyloid formation of this viral RNA-binding protein may foreshadow a much wider field for investigation.

In summary, this work extends knowledge of amyloidogenic viral proteins and their LCD segments, associates NCAP with known amyloid-forming RNA-binding proteins, and may inspire future investigation of NCAP amyloid formation in SARS-CoV-2 infection. Finally, we also suggest an approach for the development of SARS-CoV-2 therapeutics via disruption of NCAP self-assembly by targeting and capping amyloid-driving steric-zipper segments of NCAP.

## Methods

### Molecular biology reagents
Phusion HF DNA polymerase, Quick Ligase, and restriction enzymes were purchased from New England BioLabs. Custom DNA oligonucleotides were synthesized by IDT (Coralville, IA). RNA oligonucleotides, S1, S1.5, S2, and the non-specific RNA (siDGCR8-1, antisense strand) were synthesized by Horizon Discovery Biosciences.

### Computational predictions and sequence alignment
**Prediction of low-complexity sequences in the NCAP of SARS-CoV-2.** The amino acid sequence of the Nucleocapsid protein of SARS-CoV-2 (NCAP; UniProtKB[74] accession number: P0DTC9 [https://covid-19.uniprot.org/uniprotkb/P0DTC9#Sequence]) was evaluated using SEG[22] with default settings: window length = 12, trigger complexity 2.2, extension complexity 2.5. LCDs were defined by strings of at least 10 low-complexity residues. Long LCDs, such as the central NCAP-LCD, were allowed no more than five interrupting non-low-complexity residues between strings of 10 or more low-complexity residues[22].

**Prediction of LCDs in the NCAPs of various coronaviruses.** A list of coronavirus Nucleocapsid proteins was downloaded from the European Nucleotide Archive (ENA; [https://www.ebi.ac.uk/genomes/virus.html]), and protein sequences were retrieved from Uniprot [https://www.uniprot.org/]. Low complexity residues were identified using the SEG algorithm[22] with default parameters (see above). Redundant low-complexity region sequences from strains of individual viruses were removed. Low-complexity region sequences were aligned in BioEdit using the ClustalW algorithm with gap penalties set to 100 in order to avoid the insertion of gaps in the aligned sequences. Gaps consisting of hyphens in between amino acid stretches in an individual sequence represent an interrupting, non-low-complexity segment of at least 20% the length of the longest LCD in the protein rather than defined gaps in the alignment. Some of these gaps were manually made larger or smaller to achieve a more accurate alignment. Supplementary Fig. 9b is the representation of this alignment in Jalview.

**Prediction of steric-zipper forming segments.** This was done on the Nucleocapsid proteins of SARS-CoV-2 (UniProtKB[74] accession number: P0DTC9) and SARS-CoV (UniProtKB[74] accession number: P59595) using the ZipperDB algorithm[26] [https://services.mbi.ucla.edu/zipperdb/].

**Prediction of PS forming regions and context-dependent interactions.** Was performed using the Fuzdrop algorithm[28] [https://fuzdrop.bio.unipd.it/predictor] on the Nucleocapsid protein of SARS-CoV-2 (UniProtKB[74] accession number: P0DTC9 [https://covid-19.uniprot.org/uniprotkb/P0DTC9#Sequence]).

### Sequence conservation
Sequence conservation analysis was performed on the LCDs of the NCAPs of SARS-CoV and SARS-CoV-2 (UniprotKB[74] accession numbers: P59595 and P0DTC9, respectively). The sequences were aligned and colored according to conservation in Jalview.

### Construct design
Full-length SARS-CoV-2 Nucleocapsid protein gene and its fragments were PCR amplified from 2019-nCoV Control Plasmid (IDT Inc., cat. no. 10006625) and spliced with N-terminal 6xHis-SUMO tag[75] using splicing by overlap extension (SOE) technique[76]. 5′ KpnI and 3′ SacI restriction sites introduced with the flanking primers were used to ligate the resulting fragments into pET28a vector. When needed, an additional round of SOE was performed to generate internal Nucleocapsid protein deletion mutants. Construct sequences were confirmed by Sanger sequencing (Laragen, Culver City, CA). Primers used for cloning are given in Supplementary Table 4, DNA sequences and alignment of translated amino acid sequences from Sanger sequencing are given in Supplementary Figs. 10 and 11, respectively.

### Protein expression and purification
NCAP segments and full-length protein were expressed as fusions to 6xHis-SUMO (6xHis-SUMO-NCAP). Plasmids were transformed into *Escherichia coli* Rosetta2 (DE3) strain (MilliporeSigma cat. no 71-397-4) and small-scale cultures were grown at 37 °C overnight in LB with 35 µg/mL kanamycin and 25 µg/mL chloramphenicol. TB with 35 µg/µL kanamycin was inoculated with overnight starter culture at a 1:100 ratio and large-scale cultures were grown at 37 °C with 225 rpm shaking until the OD600 reached -0.6. Protein expression was induced with 1 mM IPTG and cultures were further incubated with shaking at 28 °C overnight, then harvested at 5000×g at 4 °C for 15 min. Bacterial pellets were either used right away or stored at −20 °C. Pellets from 2–4 L of culture were re-suspended in -200 mL chilled Buffer A (20 mM Tris pH 8.0, 1 M NaCl) supplemented with Halt Protease Inhibitor Cocktail (ThermoScientific cat. no. 87785) and sonicated on the ice at 80% amplitude for a total sonication time of 15 min, with pauses at regular intervals so the sample does not exceed 15 °C. Cell debris was removed via centrifugation at 24,000×g at 4 °C for 30- 60 min, filtered twice through 0.45 µm high particulate syringe filters (MilliporeSigma cat. no. SLCRM25NS), and imidazole added to 5 mM. Filtered clarified lysate was loaded onto HisTrap HP columns (GE Healthcare) and proteins were eluted over a step-gradient with Buffer B (20 mM Tris pH 8.0, 1 M NaCl, 500 mM imidazole), with extensive low-imidazole (<20%) washes to improve purity. NCAP proteins were generally eluted in 20–50% Buffer B. Fractions were analyzed by SDS−PAGE, pooled and dialyzed against 20 mM Tris pH 8.0, 250 mM NaCl at 4 °C overnight. Following dialysis, the sample was concentrated using Amicon Ultra-Centrifugal filters (MilliporeSigma) and urea was added up to 1 M final concentration if protein precipitation was observed. Ulp1 protease (homemade) was added at a 1:100–1:200 w/w ratio to purified proteins, along with 1 mM DTT, and the sample was incubated at 30 °C with 195 rpm shaking for 1–2 h. After cleavage, NaCl was added to 1 M final concentration to reduce aggregation, and the sample was incubated with HisPur Ni-NTA resin (Thermo Scientific cat. no. PI88222) equilibrated in Buffer A at 25 °C with 140 rpm shaking for 30 min. Cleaved NCAP proteins were eluted from the resin via gravity flow chromatography, then the resin was washed twice with Buffer A, twice with Buffer A + 5 mM imidazole, and finally with Buffer B. The flow-through and appropriate washes were concentrated and flash-

frozen for storage or further purified by gel filtration. Directly prior to gel filtration, the sample was centrifuged at 21,000×*g* for 30 min at 4 °C to remove large aggregates. Soluble protein was injected on a HiLoad Sephadex 16/600 S200 (for proteins larger than ~25 kDa) or S75 (for proteins smaller than ~25 kDa) (GE Healthcare) equilibrated in SEC buffer (20 mM Tris pH 8.0, 300 mM NaCl) and run at a flow rate of 1 mL/min. Elution fractions were assessed by SDS−PAGE for purity, and confirmed to have low RNA contamination as assessed by 260/280 nm absorbance ratio. Pooled fractions were concentrated and 0.2-μm filtered. Protein concentration was measured by A280 absorbance using a NanoDrop One (ThermoScientific) and calculated by the sequence-specific extinction coefficient, and aliquots were flash-frozen and stored at −80 °C. Of note, the first N-terminal residue in all purified proteins (residue #1) is a threonine remaining from cleavage of the 6xHis-SUMO tag during protein purification.

### Rosetta-based peptide inhibitor design

Crystal structures of LCD segments GSQASS and AALALL (form 1) were used as templates for the design of peptide inhibitors in Rosetta3 software[34]. 5 layers of the steric zipper structure were generated. A 6-residue peptide chain was placed at the top or bottom of the fibril-like structure. Rosetta Design was used to sample all amino acids and their rotamers on the sidechains of the fixed peptide backbone. The lowest energy conformations of the sidechains were determined by minimizing an energy function containing terms for Lennard−Jones potential, orientation-dependent hydrogen bond potential, solvation energy, amino acid-dependent reference energies, and statistical torsional potential dependent on the backbone and sidechain dihedral angles. Buried surface area and shape complementarity were scored by AREAIMOL[77] and Sc[78], respectively, from the CCP4 suite of crystallographic programs[79]. Solvation-free energy estimates were calculated using software available here: [https://doi.org/10.5281/zenodo/6321286]. Design candidates were selected based on their calculated binding energy to the top or bottom of the fibril-like structure, shape complementarity, and propensity for self-aggregation. The binding energy for an additional strand of the native sequence (i.e., AALALL) was computed for comparison with peptide inhibitor designs. The structural model of each candidate peptide was manually inspected in PyMOL[80]. Many computational designs produced sequences with high hydrophobic content, thus two arginine residues were added onto the N-terminal end to increase peptide solubility. Candidate G12 was the most effective inhibitor in preliminary screens and therefore was chosen for further evaluation.

### Peptide synthesis and purification

The NCAP steric zipper segments $_{179}$GSQASS$_{184}$ and $_{243}$GQTVTK$_{248}$ were synthesized by LifeTein. The inhibitor candidate G12 was synthesized by LifeTein and GenScript. All peptides were synthesized at over 98% purity. The NCAP segment $_{217}$AALALL$_{222}$ was synthesized and purified in-house as H-AALALL-OH. Peptide synthesis was carried out at a 0.1 mmol scale. A 2-chlorotrityl chloride resin (Advanced Chemtech) was selected as the solid support with a nominal loading of 1.0 mmol/g. Each loading of the first amino acid was executed by adding 0.1 mmol of Fmoc-Leu-OH (Advanced Chemtech FL2350/32771) and 0.4 mmol of diisopropylethylamine (DIPEA), dissolved in 10 mL of dichloromethane (DCM), to 0.5 g of resin. This mixture was gently agitated by bubbling with air. After 30 min, the supernatant was drained, and the resin was rinsed twice with 15 mL aliquots of the capping solution, consisting of 17:2:1 DCM/MeOH/DIPEA. With the first amino acid loaded, the elongation of each polypeptide was completed in a CEM Liberty Blue™ Microwave Peptide Synthesizer. A 1.0 M solution of N,N′-diisopropylcarbodiimide (DIC) in DMF was used as the primary activator, and a 1.0 M solution of ethyl cyanohydrox-yiminoacetate (oxyma) in DMF, buffered by 0.1 M of DIPEA was used as a coupling additive. The Fmoc-L-Ala-OH used was also purchased from

Advanced Chemtech (FA2100/32786). The microwave synthesizer utilizes 0.2 M solutions of each amino acid. For the deprotection of N-termini, Fmoc protecting groups, a 9% w/v solution of piperazine in 9:1 N-Methyl-2-Pyrrolidone to EtOH buffered with 0.1 M of oxyma was used. For 0.1 mmol deprotection reactions, 4 mL of the above deprotection solution was added to the resin. The mixture was then heated to 90 °C for 2 min while bubbled with nitrogen gas. The solution was drained, and the resin was washed 4 times with 4 mL aliquots of DMF. For 0.1 mmol couplings, 2.5 mL of 0.2 M amino acid solution (0.5 mmol) was added to the resin along with 1 mL of the DIC solution (1.0 mmol) and 0.5 mL of oxyma solution (0.5 mmol). This mixture was agitated by bubbling for 2 min at 25 °C, then heated to 50 °C followed by 8 min of bubbling. After the last deprotection, the resin was washed with methanol, diethyl ether, dried over the vacuum, and introduced to a cleavage cocktail consisting of 20 mL of trifluoroacetic acid (TFA), 0.50 mL of water, 0.50 mL of triisopropylsilane (TIS). After 2 h of vigorous stirring, the mixture was filtered, and the filtrate was concentrated in vacuo. The residue was triturated with cold diethyl ether, and precipitated, the crude peptide was collected by filtration. The crude peptide was then purified by RP-HPLC, using an Interchim puriFlash® 4125 Preparative Liquid Chromatography System equipped with a Luna (Phenomenex, C18(2), 5 μm, 100 Å, 30 × 100 mm) column. For purification, two buffer systems were utilized. Initial purifications and salt exchanges were executed with a 13 mM aqueous solution of trifluoroacetic acid (TFA; [A]) and a 2:3 water to acetonitrile solution, buffered by 13 mM of TFA ([B]). For the better resolution of diastereomers and other impurities, ultrapure water, buffered by 14 mM of HClO$_4$, and a 2:3 water to acetonitrile solution, buffered by 5.6 mM of HClO$_4$, were selected as mobile phases A and B, respectively. The purity of the purified fractions was analyzed by RP-HPLC, using an Agilent 1100 Liquid Chromatography System equipped with a Kinetex (Phenomenex, C18, 5 μm, 100 Å, 4.6 × 250 mm) column. Ultrapure water with 0.1% TFA, and a 1:9 water to acetonitrile solution with 0.095% TFA were selected as mobile phases [A] and [B], respectively. The flow rate was set at 1.0 mL/min and the gradient used is detailed in Supplementary Table 5. The UV absorption at 214 nm was monitored. The resulting chromatogram is shown in Supplementary Fig. 12.

### RNA in vitro transcription and purification

The nucleic acid sequence corresponding to S2hp (Supplementary Table 1) was cloned from a gBlock (IDT) of the first 1000 nucleotides of the 5′-end of the SARS-CoV-2 genome into pUC19 vectors using the restriction sites EcoRI and KpnI. Forward primer P2627 (5′-TAAT ACGACTCACTATAGGCTGTGTGGCTGTCACTCG-3′) containing the T7 promoter sequence was added at a low concentration of 0.5 nM in addition to forward primer P1471 (5′-GCGAATTCTAATACGACT CACTATAGG-3′) containing the EcoRI restriction sequence and T7 promoter sequence at the normal concentration of 500 nM. Reverse primer P2644 (5′-CGGGGTACCTCGTTGAAACCAGGGACAAG-3′) containing the KpnI restriction sequence was added at 500 nM. The clone was sequence-confirmed and the miniprep was used as a template for PCR. The forward primer for PCR containing the T7 promoter sequence was biotinylated on the 5′ end for removal of PCR template after transcription. The PCR product was purified by HiTrap column. The running buffer solutions (0.2-μm filtered) contained 2 M NaCl, 10 mM HEPES pH 7.0 (buffer A), and 10 mM NaCl, 10 mM HEPES pH 7.0 (buffer B). The purified PCR products were concentrated using Amicon Ultra centrifugal filter units (Millipore) and buffer-exchanged against 10 mM Na/HEPES pH 7.0. Transcription reactions ranging from 5 to 100 mL were set up. The transcription reaction was incubated at 37 °C with gentle shaking for one hour. After transcription, streptavidin beads (ThermoFisher) were added to the transcription and set on a rotator at room temperature for an additional 15 min. The transcription reaction was centrifuged at 500×*g* for 10 min at 4 °C. The supernatant was decanted and the pellet containing any PCR template

remaining was discarded. The transcription reaction was then purified by 5 mL HiTrap Q HP column in several rounds, loading ~5 mL into the column each round. The purified RNA was concentrated using Amicon Ultra-15 (Millipore) and the buffer was exchanged into 10 mM HEPES pH 7.0. The purity of the RNA was confirmed using denaturing poly-acrylamide gels. The concentration was calculated by measuring OD260 and a conversion factor of 40 μg/mL/OD260.

### PS assays
All solutions were prepared using DNase/Rnase-free water (ultrapure water) and were filtered twice using a 0.22-μm syringe filter. Preparations were done under sterile conditions and using sterile filter pipette tips to prevent RNA degradation.

**PS of NCAP with ThS staining (Supplementary Fig. 2a).** Experiments were carried out in 96-well black/clear glass-bottom plates (Cellvis glass-bottom plates cat. no. P96-1.5H-N). S2hp RNA, stored at −20 °C was thawed, then annealed by heating at 95 °C for 3 min and transferring quickly on the ice. The RNA was diluted by its original buffer of 10 mM HEPES pH 7.0 to 750 μM and 75 μM working solutions. 1 mM ZnCl$_2$ was prepared in ultrapure water and filtered twice with a 0.22-μm syringe filter. Fresh Thioflavin S (ThS) solution was prepared from powder (MP Biomedicals) in ultrapure water at 0.002% w/v and filtered. Purified NCAP stock solution was centrifuged at 15,000×g for 15 min at 4 °C to remove large aggregates. NCAP, S2hp vRNA, and ZnCl$_2$ were mixed in PBS at final concentrations of 30 μM NCAP with 0 or 0.75 μM S2hp vRNA, and 0 or 20 μM ZnCl$_2$ as indicated in the figure. ThS was diluted into the wells to a final concentration of 0.0002% w/v. Blank solutions containing everything but NCAP were prepared as controls. After dispensing the samples the plates were immediately covered with optical film (Corning Sealing Tape Universal Optical) and incubated in a plate reader (BMG LABTECH FLUOstar Omega) at 37 °C with 700 rpm shaking. The plates were imaged at indicated time points of incubation. All samples were imaged with ZEISS Axio Observer D1 fluorescence microscope with ZEN 2 software, equipped with a 100x oil objective lens, using the 1,4-Diphenylbutadiene fluorescence channel with a DAPI filter for ThS, as well as a DIC filter. Images were processed and rendered with FIJI (imageJ)[81].

**PS of the LCD segment with ThS staining (Fig. 3).** S2hp and ThS solutions were prepared as above. Purified LCD protein solution was centrifuged at 15,000×g for 15 min at 4 °C to remove large aggregates. The Protein, RNA, and ThS were then mixed in wells of 96-well black/clear glass-bottom plate (Cellvis glass-bottom plates cat. no. P96-1.5H-N) at 40:1 and 4:1 LCD: S2hp vRNA molar ratios in triplicates. ThS was added to 0.0002% w/v final concentration. This experiment was repeated with both 30 and 10 μM final LCD concentrations showing similar results. Respective protein and RNA blank solutions were prepared as controls. The plate was immediately covered with an optical film (Corning Sealing Tape Universal Optical) and incubated at 37 °C with 700 rpm shaking in a plate reader (BMG LABTECH FLUOstar Omega). Images were obtained at indicated time points and processed as above.

**PS of NCAP with G12 (Fig. 5).** Directly prior to assay setup, purified NCAP protein was centrifuged at 15,000×g for 15 min at 4 °C to remove large aggregates and the supernatant was used for the experiment. S2hp RNA was briefly annealed by heating at 95 °C for 3 min and transferring quickly on the ice. G12 stock solutions were prepared in DMSO in 1 mM concentration from lyophilized peptide powder and serially diluted in PBS buffer complemented with 10 % DMSO and was added to wells of 384-well black/clear glass-bottom plate containing 10 μM NCAP protein and 0.25 μM S2hp RNA (40:1 molar ratio) in PBS buffer. NCAP: G12 (or buffer control) molar ratios are indicated in the figure. The final DMSO concentration in all wells was 1%. The plate was

covered with optical film (Corning Sealing Tape Universal Optical) and incubated for ~4 h at room temperature without shaking prior to imaging. Images were acquired using an Axio Observer D1 microscope (Zeiss) with ZEN 2 software, equipped with a ×100 oil objective lens using a DIC filter. Images were processed and rendered with FIJI (imageJ)[81]. Mean area and mean circularity (weighted by particle size) of particles and droplets were calculated using MATLAB as described in the Brightfield Image Segmentation and Shape Analysis section.

**PS with FITC labeled G12 (Supplementary Fig. 6).** FITC-labeled G12 stock solution (made in DMSO) was added to non-labeled stocks at a 1:9 labeled:non-labeled ratio. The mixture was then added to a 96-well plate with glass bottom at a final concentration of 10 μM NCAP, 0.25 μM S2hp RNA (40:1 molar ratio), and 0 or 10 μM G12 in 20 mM Tris pH 8, 50 mM NaCl, and 20 mM ZnCl$_2$. The final DMSO concentration in all wells was 0.5 %. The plate was covered with optical film (Corning Sealing Tape Universal Optical) and incubated at 37 °C without shaking for 24 h prior to imaging with ZEISS Axio Observer D1 fluorescence microscope with ZEN 2 software, equipped with a ×100 oil objective lens, using the FITC fluorescence channel with a GFP filter and a DIC filter. Images were processed and rendered with FIJI (imageJ)[81].

**Measurements of ThS fluorescence in LCD PS droplets (Fig. 3c)**
The PS experiment of the LCD segment with ThS staining was performed as described above. To evaluate the change in ThS fluorescence upon incubation of the PS droplets we combined for each experimental condition and time point 5 fluorescence images per well from triplicate wells and 3 biological repeats (n = 45 images per condition per time point). Background fluorescence was subtracted individually from each image using FIJI after measuring the mean gray value and STD of a region containing no features of interest and calculating it with Eq. (1):

$$\text{Background fluorescence signal} = 3 \times \text{STD} + \text{mean gray value} \quad (1)$$

Then the mean fluorescence (gray value) of the entire background subtracted image was measured and averaged across all images from the same condition and time point. The plot was rendered in Prism software and error bars represent standard error of the mean. Two-tailed t-test with Welch's correction was performed in Prism to evaluate statistical significance of the change in ThS fluorescence between time points of each condition. Mean area and mean circularity (weighted by particle size) of particles and droplets were calculated using MATLAB as described below in the Brightfield Image Segmentation and Shape Analysis section.

**Image segmentation and Shape analysis (Figs. 3 and 5)**
Brightfield microscopy images were imported into MATLAB 9.13.0 (R2022b) where all subsequent processing and image analysis were carried out. Image segmentation was carried out by initial Gaussian filtering of each image to achieve local smoothing of the image data. Each image was filtered using a Gaussian kernel with a standard deviation of 5 pixels (px). The Laplacian of the Gaussian-filtered image was then found to highlight areas of rapid change in intensity to facilitate edge detection. Edge detection was performed on each image by finding points of maximum local gradients, using the Sobel approximation to derivatives that are implemented using the MATLAB Image Processing Toolbox. The detected edges on the resulting binary image were then dilated and holes, defined by the connectivity of edges and corners, filled. Regions with an area <100 px$^2$ were removed to reduce segmentation errors. For small regions, defined by an area less than 10,000 px$^2$, refined segmentation was then carried out in which each region was extracted from the unprocessed image data using a padded square extraction box with a side length of 1.5 times the maximal length of the region on the xy-plane. Image segmentation was

carried out on each extracted small region individually as described above, with the difference of using the Canny algorithm, implemented using the MATLAB Image Processing Toolbox, for edge detection[82]. Each detected region of the segmented image then represented an area of interest for which shape analysis was carried out. For each region, its area was found from the total number of pixels and the circularity of the area was calculated as shown in Eq. (2).

$$circularity = 4\pi \times area \times perimeter^{-2} \qquad (2)$$

For the representation of LCD assemblies with S2hp vRNA (Fig. 3), calculated circularity measures were then weighted by the area of each corresponding region in the analysis of the sample means and standard errors of the means. Statistical analysis of the area and weighted circularity of segmented regions from a total of 45 images combined from 5 individual images collected from each technical triplicate of 3 biological repeats, was finally performed and visualized as boxplots showing the 25th percentile, median, and 75th percentile of the mean values for triplicate experiments. The whiskers of the plots extend to the most extreme data points. Observations beyond the whisker length (shown as circles in the figure) are values more than 1.5 times the interquartile range beyond the bottom or top of the box. For the representation of NCAP particles with G12 (Fig. 5), calculated circularity measures were weighted by the area of each corresponding region in the analysis of the sample means. Mean area and mean weighted circularity was calculated across regions of 15 images per experimental condition, obtained by combining 5 images for each technical triplicate. Every biological repeat was analyzed separately. A representative boxplot is shown in the figure, in which the central mark indicates the median of the experimental triplicate means, and the bottom and top edges of the box indicate the 25th and 75th percentiles, respectively.

### Thioflavin-T assays
All solutions in these experiments were prepared using DNase/RNase-free water (ultrapure water) and were filtered twice using a 0.22-μm syringe filter. Preparations were done under sterile conditions and using sterile filter pipette tips to ensure RNA preservation. Thioflavin T (ThT) stock solution was freshly prepared from powder (Sigma, CAS ID: 2390-54-7) at a concentration of 20 mM in DNase/RNase ultrapure water, followed by 0.22-μm filtration.

**Thioflavin-T fibrillation kinetic assays.** Purified NCAP protein and its segments were separately diluted into 20 mM Tris pH 8.0, 300 mM NaCl buffer at 235 μM concentration. S2hp RNA was diluted by 10 mM HEPES pH 7.0 buffer to 75 μM concentration. The proteins, RNA and ThT were then mixed to final concentrations of 300 μM ThT, 30 μM protein, and 0 or 7.5 μM RNA (as indicated in Fig. 1 and Supplementary Fig. 2), in 1X PBS pH 7.4. Blank samples containing everything but the protein were prepared. The reaction was carried out in a black 384-well clear-bottom plate (NUNC 384) covered with optical film (Corning Sealing Tape Universal Optical) and incubated in a plate reader (BMG LABTECH FLUOstar Omega) at 37 °C, with 700 rpm double orbital shaking for 30 s before each measurement. ThT fluorescence was measured with excitation and emission wavelengths of 430 and 485 nm, respectively. Measurements were made with technical triplicates for each sample. All triplicate values were averaged, and blank readings from samples without proteins were averaged and subtracted from the values of corresponding protein mixtures. The results were plotted against time. The experiment was repeated at least three times on different days.

**Thioflavin-T endpoint assay.** Purified LCD protein segment was diluted into 20 mM Tris pH 8.0, 300 mM NaCl buffer at 235 μM concentration. The proteins and ThT were then mixed to final

concentrations of 300 μM ThT and 100 μM protein, in 1X PBS pH 7.4. A blank sample containing everything but the protein was prepared and measured as a buffer control. Fibril formation was carried out in parafilm-covered PCR tubes, incubated in a floor shaker (Torrey Pines Scientific Inc, Orbital mixing chilling/heating plate) at 37 °C, with fast mixing speed for 11 days. 30 μL of the samples were taken out of the tubes at days 1, 6, and 11of incubation and put in a black 384-well clear-bottom plate (NUNC 384) covered with optical film (Corning Sealing Tape Universal Optical) and incubated in a plate reader (BMG LAB-TECH FLUOstar Omega) at 37 °C, with 700 rpm double orbital shaking for 30 s before the measurement. ThT fluorescence was measured with excitation and emission wavelengths of 430 and 485 nm, respectively.

### Turbidity assay
All solutions were prepared using DNase/RNase-free water (ultrapure water) and were filtered twice using a 0.22-μm syringe filter. Preparations were done under sterile conditions and using sterile filter pipette tips to ensure RNA preservation. Protein and RNA working solutions were prepared as described above for the ThT experiment of NCAP and its segments. Each reaction sample contained 30 μM protein and 0 or 7.5 μM RNA in 1X PBS pH 7.4. Blank samples contained everything but the protein. The reaction was carried out in a black 384-well clear-bottom plate (NUNC 384) covered with optical film (Corning Sealing Tape Universal Optical) and incubated in a plate reader (BMG LAB-TECH FLUOstar Omega) at 37 °C, with mixing before and between measurements. Turbidity was measured with absorbance (OD) at 600 nm. Measurements were made with technical triplicates for each sample. Triplicate values were averaged, and appropriate blank readings (samples without the protein) were averaged and subtracted from the corresponding readings. The results were plotted against time. The experiment was repeated at least three times on different days.

### Negative stain transmission electron microscopy (TEM)
Samples for negative staining TEM were prepared as described below. All solutions in these experiments were prepared using DNase/RNase-free water (ultrapure water) and were filtered twice using a 0.22-μm syringe filter. Preparations were done under sterile conditions and using sterile filter pipette tips to ensure RNA preservation. For grid preparation and screening, 4 μL of each sample was applied directly onto 400-mesh copper TEM grids with Formvar/Carbon support films (Ted Pella), glow discharged (PELCO easiGlowxs) for 45 s at 15 mA immediately before use. Grids were incubated with the samples for 2 min, then the samples were blotted off using filter paper. The grids were washed three times with water and once with 2% uranyl acetate solution with blotting after each wash. The grids were then incubated with 6 μL of uranyl acetate solution for 30–45 s before blotting. Micrographs were imaged using an FEI Tecnai T12 microscope at room temperature with an accelerating voltage of 120 kV. Images were recorded digitally with a Gatan US 1000 CCD camera, using the Digital-Micrograph® Suite software, and processed in the ImageJ[83] software.

**NCAP fibrils from PS droplets formed in PBS.** NCAP samples with and without 0.75 μM S2hp and 20 μM ZnCl₂ (Supplementary Fig. 2b) were prepared in PBS as described in the PS method section. Samples were vigorously scrapped from the bottom of the wells after 6 days of incubation using a 100 μl pipette tip and used for TEM grid preparation. A blank control containing 0.75 μM S2hp, 20 μM ZnCl₂ and 0.0002% w/v ThS in PBS was imaged as well.

**NCAP fibrils in 2 mM Tris pH 8.0, 30 mM NaCl (Supplementary Fig. 2d).** Purified NCAP was diluted to 50 μM final concentration from its stock solution (made in 20 mM Tris pH 8.0, 300 mM NaCl buffer) into ultrapure water supplemented with ZnCl₂ in 0 (water only) or 20 μM final concentration. Samples were incubated for 3 days with acoustic resonance mixing at 37 °C using a custom-built 96-well plate

shaker set to 40 Hz. The samples were then recovered and applied to the EM grid as described above.

**Fibrils of NCAP and its segments in PBS.** NCAP (Supplementary Fig. 2c) and its segments (Fig. 1f) were separately diluted to 235 μM concentration by 20 mM Tris pH 8.0, 300 mM NaCl. The S2hp RNA was diluted to 250 μM by 10 mM HEPES, pH 7.0 buffer. The proteins and RNA were further diluted in 1X PBS pH 7.4 such that each reaction sample contained 100 μM protein and 0/ 25 μM RNA. Fibril formation was carried out in parafilm-covered PCR tubes, incubated in a floor shaker (Torrey Pines Scientific Inc, Orbital mixing chilling/heating plate) at 37 °C, with fast mixing speed for 6 (LCD and DD-C$_{term}$) to 14 (NCAP) days.

**LCD fibril formation with short RNA segments (Fig. 2b).** RNA stock solutions were thawed, then annealed by heating at 95 °C for 3 min and transferring quickly on the ice. The RNAs were diluted to 1 mM concentration by their original buffer of 10 mM HEPES pH 7.0. LCD protein stock was freshly thawed and added together with the appropriate RNA solution into 1X PBS to reach a 1:2 protein:RNA molar ratio at 50 μM final concentration of LCD, in 50 μL final volume in a black 384-well clear-bottom plate (NUNC 384). The plate was covered with optical film (Corning Sealing Tape Universal Optical) and incubated in a plate reader (BMG LABTECH FLUOstar Omega) at 37 °C with shaking. Samples were taken for TEM screening after 12 h (day 1) and 4 days of incubation.

### X-ray fiber diffraction

**LCD with and without S2hp vRNA.** 1.27 mM purified LCD stock solution was thawed and dialyzed in a dialysis cassette with a 3.5 kDa cutoff (Thermo Scientific cat. no. 87724) for 4 h at RT in 20 mM Tris pH 7.4, 50 mM NaCl buffer with or without the addition of S2hp vRNA in 4:1 LCD:S2hp molar ratio (955 μM protein and 236 μM RNA). After dialysis, the samples were added to a black 384-well clear-bottom plate (NUNC 384), covered with optical film (Corning Sealing Tape Universal Optical), and incubated in a plate reader (BMG LABTECH FLUOstar Omega) at 37 °C, with 30 s of 700 rpm double orbital shaking every 5 min for 3 weeks. The fibrils were pelleted and washed three times in water by centrifugation at 13,000×g for 10 min at RT, then pelleted again and resuspended in 5 μL of deionized water. Fibrils were aligned by pipetting 2 μL of the fibril resuspension in a 3 mm gap between two fire-polished glass rods, positioned end-to-end. After 1 h of drying at room temperature, another 2 μL of the fibril suspension was applied, thickening the sample. After another hour of drying, the aligned fibril sample was transferred to the exterior of a standard crystal mounting loop. To glue the sample to the loop, the loop was wetted with 50 % v/v ethylene glycol solution, then touched to the surface of the sample and immediately plunged in liquid nitrogen. The samples were shipped to the Advanced Photon Source, beamline 24-ID-E at Argonne National Laboratory for remote data collection. The sample was kept at 100 K using a nitrogen cryo-stream. Diffraction patterns were collected on a Dectris Eiger 16M pixel detector using a 2 s exposure at 100% transmission and 1-degree rotation. The X-ray beam wavelength was 0.9792 Å and impinged on the sample only, avoiding the loop and ethylene glycol, so these later materials do not contribute to the diffraction pattern. The detector was placed 350 mm from the sample. Diffraction images were displayed with the ADXV program (Scripps).

**LCD with non-specific RNA (antisense siDGCR8-1 RNA).** LCD stock solution was concentrated to 2.2 mM and the buffer was exchanged to 20 mM Tris pH 8.0, 150 mM NaCl in a centrifugal filter with 3 kDa cutoff (Milliepore Sigma Amicon Ultra cat. no. C82301). Antisense siDGCR8-1 RNA stock solution, stored at −20 °C, was thawed and combined with the LCD solution in 1:3 LCD:RNA molar ratio. The solution was titrated to reach a final pH of ~5 as confirmed with pH paper. The final protein

concentration was 283 μM and RNA concentration was 849 μM. The reaction mixture was incubated in a floor shaker (Torrey Pines Scientific Inc, Orbital mixing chilling/heating plate) at 37 °C, with rapid mixing speed for 7 days. The fibrils were prepared and mounted as described above except that the fibrils were aligned with a single application of 5 μL of the fibril suspension, rather than two smaller applications. Diffraction was measured at beamline24-ID-C, rather than 24-ID-E. Diffraction patterns were collected on a Dectris Eiger2 16M pixel detector using a 1 s exposure at 90% transmission and 0.5-degree sample rotation. The X-ray beam wavelength was 0.9791 Å and impinged on the sample only, avoiding the loop and ethylene glycol, so these later materials do not contribute to the diffraction pattern. Exposures were collected at sample-to-detector distances of 200 and 500 mm. Diffraction images were displayed with the ADXV program (Scripps).

### Crystallization of NCAP peptide segments

The NCAP segment $_{217}$AALALL$_{222}$ crystallized in batch just before the purification by RP-HPLC. The peptide had been deprotected and cleaved from the resin, triturated with cold diethyl ether, and precipitated. Most of the product had been collected via filtration, but some residual peptide remained in the round bottom flask and we intended to use this residual peptide to check the peptide purity by analytical HPLC. We dissolved the residual peptide with water, acetonitrile, and TFA in a volume ratio of approximately 45:45:10 and transferred it to a 1 mL glass vial for HPLC injection. The solution was left in the sample holder and needle-like crystals formed after a week. Some of these crystals were retained for crystal structure determination. The bulk of the peptide was further purified, as described above. Later, we showed we could reproduce the crystals by dissolving 0.75 mg of AALALL in 50 μL of TFA and then diluting with 225 μL of acetonitrile and 225 μL of water. This was left to sit in an HPLC vial which had its septum top poked open once with the HPLC injection needle, and the same crystal form appeared in 3 months.

We screened for additional AALALL crystals using 96-well kits and purified peptide dissolved at 10 mg/mL concentration in 19.6 mM LiOH. Crystals were grown by the hanging drop vapor diffusion method. The UCLA Crystallization Facility set up crystallization trays with a Mosquito robot dispensing 200 nL drops. Needle-shaped crystals of $_{217}$AALALL$_{222}$ grew at 20 °C in a reservoir solution composed of 30% w/v polyethylene glycol (PEG) 3000 and 0.1 M n-cyclohexyl-2-aminoethanesulfonic acid (CHES), pH 9.5. The purified NCAP segment $_{179}$GSQASS$_{184}$ was dissolved in water at 100 mg/mL concentration. Hanging drop crystallization trays were set using 200 nL drops. Needle-shaped crystals grew at 20 °C using a reservoir solution composed of 1.0 M Na, K tartrate, 0.2 M Li$_2$SO$_4$, and Tris pH 7.0. Needle-shaped crystals appeared immediately after setting up the tray. The purified NCAP segment $_{243}$GQTVTK$_{248}$ was dissolved in water at 68 mg/mL concentration. Hanging drop crystallization trays were set using 200 nL drops. Needle-shaped crystals appeared within 1 day at 20 °C using a reservoir solution composed of 2.0 M (NH$_4$)$_2$SO$_4$, 0.1 M sodium HEPES, pH 7.5, and 2% v/v PEG 400.

### Structure determination of NCAP peptide segments

Microfocus X-ray beam optics were required to measure crystal diffraction intensities from our crystals since they were needle-shaped and less than 5 microns thick. We used microfocus beamline 24-ID-E of the Advanced Photon Source located at Argonne National Laboratory. Crystals were cooled to a temperature of 100 K. Diffraction data were indexed, integrated, scaled, and merged using the programs XDS and XSCALE[84]. Data collection statistics are reported in Table 1. Initial phases for AALALL and GSQASS were obtained by molecular replacement with the program Phaser[85] using a search model consisting of an ideal β-strand with sequence AAAAAA. Phases for GQTVTK were obtained by direct methods using the program ShelxD[86]. Simulated annealing composite omits maps[57] were calculated using Phenix[87].

Refinement was performed using the program Refmac[88]. Model building was performed using the graphics program Coot[89]. Structure illustrations were created using PyMOL[80]. Residue hydrophobicity of the steric zipper segments was assigned and colored according to the Kyte and Doolittle hydrophobicity scale embedded in UCSF Chimera[90].

**G12 evaluation in HEK293-ACE2 cells infected with SARS-CoV-2**
Lyophilized G12 peptide powder was dissolved in 100 % DMSO (Sigma cat. no. D2650) to approximately 10 mM, centrifuged at $21,000 \times g$ for 30 min to remove large aggregates, then aliquoted and stored at −20 °C until use. To determine peptide concentrations accurately, the stock was diluted in UltraPure distilled water (ThermoFisher cat. no. 10977015), and the concentration was measured using the Pierce Quantitative Fluorometric Peptide Assay (ThermoFisher cat. no. 23290). HEK293-ACE2 cells (ATCC, cat. no. CRL-3216, authenticated and quality tested by ATCC [https://www.atcc.org/products/crl-3216]) stably over-expressing the human ACE2 receptor[91] were cultured in DMEM (Gibco cat no. 11995-065) supplemented with 10% FBS (Gibco cat no. 26140-079), 1% penicillin-streptomycin (Gibco cat no. 15140-122), 10 mM HEPES pH 7 (Gibco cat no. 15630106), 50 µM 2-mercaptoethanol (Sigma cat no. M3148), and 1 µg/mL puromycin (Gibco cat no. A1113803) for selection, at 37 °C, 5 % $CO_2$. Cells were confirmed negative for mycoplasma by PCR using a Universal Mycoplasma Detection Kit (ATCC cat. no. 30-1012K). The HEK293-ACE2 cells were plated in 96-well black/clear plates (Greiner Bio-One cat. no. 655090) at $2 \times 10^4$ cells per well. The cells were incubated for 1–2 days at 37 °C, 5% $CO_2$, then exchanged into antibiotic-free media and incubated for an additional day. Cells were then transfected with the peptide-based inhibitors, either unlabeled (Fig. 5d and Supplementary Fig. 8; Final peptide concentrations are detailed in the figures), or with ~15 µM of FITC-labeled G12 (Supplementary Fig. 7) by diluting stock solutions (made in 5% DMSO) into cell culture medium to a 10X concentration, and serially diluting from there for dose–response assays while maintaining similar DMSO concentration in all peptide dosages (Fig. 5d and Supplementary Fig. 8). 10 µL of 10X peptide diluted in culture medium was added to 90 µL media in each well, for a final DMSO concentration of 0.5% in all wells. Finally, Endo-Porter (PEG-formulation) transfection reagent (GeneTools LLC, Philomath, OR) was added to each well at a final concentration of 6 µM. Plates were incubated for 2- 4 h, then the cells were infected with SARS-CoV-2 (Isolate USA-WA1/2020) (BEI Resources) in the UCLA BSL3 High-Containment Facility[91] by adding the virus in 200 µl final volume at an MOI of 0.05 for evaluation of dose dependence antiviral activity with the inhibitor G12 (Fig. 5d and Supplementary Fig. 8). The uninfected control received only the base media used for diluting the virus. The plates were incubated for an additional 24 h at 37 °C, 5% $CO_2$, and fixed with 100% methanol for immunofluorescence assay. Fixed cells were washed 3 times with PBS pH 7.4 (Gibco cat. no. 10010-023) and incubated with blocking buffer (2% BSA, 0.3% Triton X-100, 5% goat serum, 5% donkey serum, 0.01% $NaN_3$ in PBS) for 2 h at room temperature. Anti-Spike protein primary antibody was diluted into blocking buffer and incubated overnight at 4 °C. Either of these primary anti-Spike protein antibodies was used (depending on availability): BEI Resources, NIAID, NIH rabbit monoclonal Anti-SARS-Related Coronavirus 2 Spike Glycoprotein S1 Domain (produced in vitro), cat. no. NR-53788, clone no. 007, Lot: HA14AP3001 (purchased from SinoBiological, cat. no. 40150-R007), at a 1:100 dilution ratio, or BEI Resources, NIAID, NIH: Mouse Monoclonal Anti-SARS-CoV S Protein (Similar to 240C), cat. no. NR-616, Lot: 102204 (purchased from ATCC), at a 1:300 dilution ratio. Following overnight incubation, cells were washed with PBS and incubated for one hour at room temperature with AlexaFluor-555 conjugated secondary goat anti-mouse (Abcam cat. no. ab150114, Lot: GR299321-5), or goat anti-rabbit (Abcam cat. no. ab150078, Lot: GR302355-2) antibody, diluted at 1:1000. All antibodies used in this section were validated by their respective vendors. Following

incubation with the secondary antibody, the cells were stained with 10 µg/mL DAPI (ThermoFisher cat. no. D1306) for 10 min, and stored in PBS for imaging. Plates were imaged using an ImageXpress Micro Confocal High-Content Imaging System (Molecular Devices, San Jose, CA) in widefield mode at 10X magnification. 9 sites per well were imaged, and the percentage of infected cells was quantified using the MetaXpress multiwavelength cell scoring module. We considered spike protein-expressing cells as infected and calculated their percentage from the total number of cells in the well. Raw values were exported into Microsoft Excel, and percent-infected cells were then normalized to an infected culture that was treated with vehicle only. Statistical analysis was performed using one-way ANOVA in GraphPad Prism, and $IC_{50}$ values were estimated (Fig. 5d) using a four-parameter non-linear fit dose–response curve.

**Cytotoxicity assay in HEK293-ACE2 cells (Fig. 5d)**
HEK293-ACE2 cells were plated and transfected with peptides following the same protocol as used for the viral assays, but following transfection were incubated at 37 °C and 5% $CO_2$ for 24 h. Peptide cytotoxicity was then assessed using the CyQUANT LDH Cytotoxicity Assay (ThermoFisher cat no. C20300) following the manufacturer protocol. Absorbance was measured at 490 and 680 nm (background subtraction) using a SpectraMax M5 (Molecular Devices) with Soft-MaxPro v5.3 software.

**Statistics and reproducibility**
All turbidity and ThT fibrillation kinetic experiments were repeated three independent times with technical triplicates. Technical triplicates were averaged and blank subtracted. Representative curves are presented in the figures. Endpoint ThT measurements of the LCD-only segment were done using three samples. Each sample was measured once per every time point. X-ray diffractions of LCD only and LCD+ S2hp fibrils were each collected three times on different days, using different diffractometers and x-ray sources while showing similar results. Diffraction of LCD+ non-specific RNA fibrils was collected twice from different regions of the same loop, showing similar results. EM micrographs of LCD-only fibrils were captured at least five independent times. LCD fibrils with the different vRNA segments were visualized by EM at least two independent times per vRNA type, once of which with different time points. NCAP with and without S2hp vRNA in PBS was imaged by EM from two independent samples. Other EM images were taken from a single sample. PS of the LCD-only segment with ThS and PS of NCAP with and without different concentrations of G12 were each performed three independent times with technical triplicates showing similar results. PS of NCAP with ThS was repeated twice (2nd repeat incubated for 3 days only) showing ThS partitioning into NCAP's PS droplets. FITC-labeled G12 was tested on NCAP PS droplets in vitro once. Antiviral activity of G12 in cells was tested three independent times with G12 concentrations of over 10 µM showing inhibition of ~40–60% in viral infectivity. Full dose response of G12 and its cytotoxicity in cells was tested in triplicated wells. Distribution of FITC labeled G12 in HEK293-ACE2 cells was tested two independent times with duplicated wells.

**Reporting summary**
Further information on research design is available in the Nature Portfolio Reporting Summary linked to this article.

## Data availability
Atomic coordinates that support the findings of this study are available in the RCSB Protein Data Bank (PDB) under accession numbers: 7LV2 [https://doi.org/10.2210/pdb7LV2/pdb], 7LTU [https://doi.org/10.2210/pdb7LTU/pdb] (form 1), 7LUX [https://doi.org/10.2210/pdb7LUX/pdb] (form 2), and 7LUZ [https://doi.org/10.2210/pdb7LUZ/pdb]. The amino acid sequences of the Nucleocapsid

proteins of SARS-CoV-2 and SARS-CoV analyzed in this study are available on UniProtKB, accession numbers: P0DTC9, and P59595 respectively. Amino acid sequences of other coronavirus Nucleocapsid proteins were accessed from the European Nucleotide Archive [ENA; https://www.ebi.ac.uk/genomes/virus.html]. Raw EM images, light and fluorescence microscopy images and fiber diffraction source files generated in this study have been deposited in the Figshare respiratory at [https://figshare.com/projects/Low_Complexity_Domains_of_the_Nucleocapsid_Protein_of_SARS-CoV-2_Form_Amyloid_Fibrils/162391]. Data for all plots presented in this manuscript are provided with this paper in the Source Data file. Source data are provided with this paper.

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

## Acknowledgements

We thank Megan Bentzel, Jose Rodriguez, Meytal Landau, and Mark Arbing for the discussions. We thank the staff at the Northeastern Collaborative Access Team, which is funded by the National Institute of General Medical Sciences from the National Institutes of Health (P30 GM124165). The Eiger 16M detector on the 24-ID-E beamline is funded by an NIH-ORIP HEI grant (S10OD021527). The Advanced Photon Source, a U.S. Department of Energy (DOE) Office of Science User Facility operated for the DOE Office of Science by Argonne National Laboratory under Contract No. DE-AC02-06CH11357. Some of this work was also performed at the Stanford-SLAC Cryo-EM Center (S2C2), which is supported by the National Institutes of Health Common Fund Transformative High-Resolution Cryo-Electron Microscopy program (U24 GM129541). The content is solely the responsibility of the authors and does not necessarily represent the official views of the National Institutes of Health. The authors also acknowledge the use of instruments at the Electron Imaging Center for NanoMachines supported by NIH (1S10RR23057 to ZHZ) and CNSI at UCLA. Mass spectrometry data were collected on instrumentation maintained and made available through the support of the UCLA Molecular Instrumentation Center—Mass Spectrometry Facility in the Department of Chemistry. This material is based upon work supported by the National Science Foundation under Grant No. (MCB 1616265), NIH/NIA R01 Grant AG048120, the U.S. Department of Energy (DOE) Contract No. DOE-DE-FC02-02ER63421, and by UCLA David Geffen School of Medicine—Eli and Edythe Broad Center of Regenerative Medicine and Stem Cell Research Award Program, Broad Stem Cell Research Center (BSCRC) COVID 19 Research Award (OCRC #20-73). This study is also supported by the UCLA W.M. Keck Foundation COVID-19 Research Award and National Institute of Health awards 1R01EY032149-01, 5U19AI125357-08, 5R01AI163216-02 and 1R01DK132735-01 to V.A. The Human Frontiers Science Project Organization (HFSPO) (LT000623/2018-L) supported E.T.-F. NIH NIGMS GM123126 grant supported Luk.S. C.-T.Z. was funded by the UCLA Dissertation Year Fellowship.

## Author contributions

Constructs design and cloning: P.M.S., Luk.S. Protein preparation and experimentation: E.T-F, J.T.B., S.L.G., X.C., R.A., J.L., Y.X.J. RNA preparation and experimentation: C.E.T., Y.L., Luk.S. Peptide preparation: C.-T.Z. X-ray crystallography: M.R.S., C.-T.Z, J.L., K.H., G.F., D.C. Fluorescence and electron Microscopy: E.T.-F., J.T.B., X.C., D.R.B., R.A., Y.X.J., H.P., G.M.R., J.L. Computational analysis and peptide self-assembly modulators design: G.M.R., P.M.S., Y.X.J., E.T.-F., Lor.S., K.A.M. Brightfield image segmentation and shape analysis in MATLAB: L.L. In-cell assays: J.T.B., G.G. Jr. Writing and figure preparation: E.T.-F., M.R.S., J.T.B., F.G., D.S.E. Technical support: D.H.A., Project management: E.T.-F., M.R.S., R.D., V.A., F.G., and D.S.E.

## Competing interests

D.S.E. is an advisor and equity shareholder in ADRx, Inc. The remaining authors declare no competing interests.
