## [Peer Review File · Nature Communications]

REVIEWER COMMENTS

Reviewer #1 (Remarks to the Author):

The authors have performed a elegant and well designed study that shows that fragments of the NCAP protein readily forms amyloid fibrils both in the presence and absence of vRNA. Further they convincingly show that at least one LCD region is required for this amyloidogenic self-assembly, suggesting a role of amyloid self-assembly in PS and RNA replication in the viral replication cycle. The authors go on to design a complimentary peptide designed to bind to the amyloidogenic regions of the LCD in order to hinder amyloid formation which may go on to affect RNA replication in SARS-CoV-2. Using an in vitro study they show that this peptide can reduce SARS-CoV-2 infection without introducing toxicity to the cell line.

I have a few suggestions which I believe should be addressed before the publication is considered for publication in Nature Communications.

1. The authors should add some discussion as to which RNA sequences are required to promote amyloid formation, and if it is specific to particular sequences or just a general non-specific binding of the negatively charged RNA? Do the same sequences bind to the NCAP and LCD regions? Does this create competition between RNA binding to the NCAP and LCD regions. If this is the case it would seem that this might reduce the efficiency of both PS and viral replication, which seems like a oddly developed evolutionary trait.

2. The ability of the full NCAP protein to form amyloid fibrils is not as convincingly proven as it is with the LCD domains.

a. The EM images of the fibrils in NCAP figure 1D are unconvincing (especially the +vRNA), I would suggest these are replaced with better images that more clearly show the presence of amyloid-like fibrils.

b. In figure 1c the NCAP +vRNA curve looks strange for a

relatively large protein due to the lack of lag phase. To me it looks much more like the aggregation of a small amyloidogenic peptide. The authors should provide an explanation as to why the ThT curve of NCAP + vRNA looks so different to LCD + vRNA. Also why do none of the proteins or fragments show +ve ThT signals in the absence of vRNA, the EM images of LCD -vRNA seem to show a high abundance of fibrils, you would expect this to result in some ThT signal?

c. In figure 3 fiber diffraction is only shown for the LCD segment, were the authors able to detect the 10 and 4.7 Å reflections from the full NCAP protein? If so this should be shown, if not some explanation should be provided.

I believe this improved characterization of NCAPs fibrillation propensity is important as the antiviral activity experiments in figure 5 are based on the full NCAP protein. Clearer images of fibril formation in Figure 5 would make it easier to argue that the G12 peptide is disrupting NCAP's ability to form amyloids, which is then going on to prevent PS (the DIC images very clearly show this) and reduce viral infection.

Reviewer #2 (Remarks to the Author):

The manuscript "Low Complexity Domains of the Nucleocapsid Protein of SARS-CoV-2 Drive Amyloid Formation" by Fligelman...Eisenberg describes a structure-function analysis of the SARS-CoV-2 nucleocapsid (NCAP) protein, in particular identifying and characterizing the proteins' propensity to form amyloid fibrils. NCAP is highly abundant in SARS-CoV-2 cells, is highly immunogenic, and is known to play crucial roles in both the production of viral RNAs and their packaging into virions. Because of its importance to different life cycle stages and its immunogenicity, NCAP has recently gained more attention as a possible target for next-generation SARS-CoV-2 vaccines and antiviral therapies.

In this paper, Fligelman et al. follow up a number of studies that have demonstrated that NCAP forms phase separated condensates both on its own and with viral (and other) RNAs. They show that the central unstructured domain (termed LCD here) is principally responsible for this behavior, and critically this is the first report to show that this region promotes the formation of amyloid fibrils in addition to less-specific condensates demonstrated by others. Drawing on their expertise in this area, they determine high-resolution 3D structures of three short regions in the LCD that form fibrils. Based on these structures, they design a peptide inhibitor of NCAP self-assembly and demonstrate that it has measurable antiviral activity in a cell culture model of SARS-CoV-2 infection.

Overall, this paper is important and timely, the data and presentation are solid and clear, and I recommend publication with only minor changes, detailed below.

There are several points in the manuscript which would benefit from more explanation. One such point is the computational analysis of NCAP described in the first sentence of the Results section. How was this analysis done? What features is the analysis looking for? How successful is this type of analysis in accurately identifying regions that undergo phase separation/amyloid formation?

Next, the manuscript would benefit from more explanation of the choice of viral RNAs used in the study. Late in the manuscript, the authors refer to a related manuscript in bioRxiv (which is itself very interesting) that details the dissection of the 5' end of the viral RNA and its interactions with NCAP. A

few sentences describing this work and how it affected the choice of RNAs for this study would help the reader.

Figure 1A: What do the capital letters (RM...ESKMS) represent in the sequence of the LCD? Are these residues that are mutated in different variants? Also on this figure panel, the authors may consider labeling/underlining the six-residue segments they identify as contributing to amyloid fibril formation.

In the discussion, the authors should address the question of NCAP phosphorylation: where are the identified phosphorylation sites relative to the identified amyloid-forming segments? How do the authors envision phosphorylation to alter the self-assembly properties of NCAP?

In a related point, there are a huge number of SARS-CoV-2 genome sequences available, and those of a number of related coronaviruses as well. How well-conserved are the identified self-associating segments within SARS-CoV-2 isolates? Are any of these regions mutated in more recently-evolved variants? Related to this, are there regions of related coronavirus N proteins that are predicted to drive similar self-assembly? This would be expected if the amyloid fiber formation were to directly contribute to viral RNA packaging.

Reviewer #3 (Remarks to the Author):

The manuscript by Tayeb-Fligelman et al. reported that the Nucleocapsid protein (NCAP) of SARS-CoV-2 can form amyloid fibrils, and inhibiting amyloid formation of NCAP by the designed peptide can diminish SARS-CoV-2 infection. Amyloid fibrillation of different proteins has been observed in different human tissues (e.g., brain, heart and skin), which is closely related to various human diseases. In this study, the authors found that NCAP of SARS-COV-2 also exhibits the capability of forming amyloid fibrils, which is an important finding. They further demonstrated that the central low-complexity domains (LCD) of NCAP can undergo liquid-like phase separation to form Thioflavin-S positive droplets and then transit to solid-like amyloid fibrils which exhibit typical intermolecular β -sheet packing features. Furthermore, the authors determined the atomic structures of three adhesive segments in the central LCD. Importantly, the authors designed a peptide which can interfere with the self-assembly of NCAP and exhibits dose-dependent antiviral activity in the cell model, which confirms the potential role of NCAP fibrillation in virus infection and on the other hand suggests a new antiviral strategy of targeting NCAP fibrillation. Overall, this work revealed the important role of the LCD of NCAP in driving its phase separation and liquid-to-solid phase transition, and indicated the potential of inhibiting the fibrillation process of NCAP for preventing virus infection. Whereas, it remains unclear whether the fibrillar form of NCAP indeed exists during virus infection and replication. To strengthen this work, the authors need to address the concerns listed below:

Major concerns:

1. Due to the poor quality and low resolution of the NS-TEM images (e.g., Nterm-RBD+vRNA in Figure 1D; NCAP+S2hp in Figure S2B iv.), it's hard to clearly see the details of the sample, and it's insufficient to distinguish whether these samples formed amorphous aggregates or amyloid fibrils. The authors need to provide NS-TEM images with high quality.
2. Figure 1D presents NS-TEM images of samples after incubation of 2 weeks to support the accelerating activity of vRNA in NCAP amyloid fibril formation. While, Figure 1C shows the results of ThT fibril formation kinetic assay within 33 hours, and the ThT curve reaches the peak within 6-8 hour and keeps decreasing after then. The authors need to address this. Whether NCAP forms fibrils when ThT intensity peaks. Moreover, from the TEM images (Fig.1D), the NCAP+vRNA sample doesn't form typical amyloid fibrils, they look more like oligomers. Thus, the authors may need to provide additional evidence to validate that NCAP indeed forms typical amyloid fibrils rather than certain types of oligomers or precipitates.
3. It's interesting that the authors report the phase separation and transition process of LCD in the presence of different molar ratio of S2hp by using different characterizing methods (Fig.2). To ensure that LCD eventually condenses into amyloid fibrils from droplet, the authors need to provide TEM images for each time point, especially the end point, where LCD droplets mostly disappear and ThS-positive puncta massively forms.
4. The authors demonstrated the roles of viral RNA in accelerating the fibril formation process of central LCD and NCAP. While, the mechanism of vRNA recognition, binding to NCAP and promoting the self-assembly process remains unclear. The authors may need to discuss this point to strengthen the interesting finding of this work.
5. Increasing concentrations of G12 were found to disrupt the phase separation process but promote the formation of larger aggregates as shown in Figure 5. This raises a question: are these larger aggregates ThS-positive amyloid aggregates? The authors may need to address this by carefully characterizing the nature of these aggregates.
6. Lower concentrations of G12 (0.5 μ M, 1.5 μ M, 4 μ M) shows an increasing relative infected cell rate (> 100%) in Figure S7 and this data needs to be explained further.

7. To what extent do fibrils from the central LCD resemble those formed by NCAP with vRNA? The author may need to discuss.

8. The incubation time of NCAP with and without ZnCl₂ in Figure S2C was not provided.

Minor concerns:

1. In the RESULTS section, the word “PS” on line 121 used an abbreviation but the full name “PHASE SEPARATION” was used on line 204.

2. The word “DROPLES” on line 121 should be “DROPLETS”.

RE: Low Complexity Domains of the Nucleocapsid Protein of SARS-CoV-2 Drive Amyloid Formation.

We thank the reviewers for their insights and suggestions for improving our manuscript. Below we address each point raised by the reviewers. Additional description of formatting-related and other general edits is provided at the end of this document. All edits are highlighted with green in the revised manuscript document.

Reviewer #1 (Remarks to the Author):

The authors have performed a elegant and well designed study that shows that fragments of the NCAP protein readily forms amyloid fibrils both in the presence and absence of vRNA. Further they convincingly show that at least one LCD region is required for this amyloidogenic self-assembly, suggesting a role of amyloid self-assembly in PS and RNA replication in the viral replication cycle. The authors go on to design a complimentary peptide designed to bind to the amyloidogenic regions of the LCD in order to hinder amyloid formation which may go on to affect RNA replication in SARS-CoV-2. Using an in vitro study they show that this peptide can reduce SARS-CoV-2 infection without introducing toxicity to the cell line.

I have a few suggestions which I believe should be addressed before the publication is considered for publication in Nature Communications.

1. The authors should add some discussion as to which RNA sequences are required to promote amyloid formation, and if it is specific to particular sequences or just a general non-specific binding of the negatively charged RNA? Do the same sequences bind to the NCAP and LCD regions? Does this create competition between RNA binding to the NCAP and LCD regions. If this is the case it would seem that this might reduce the efficiency of both PS and viral replication, which seems like a oddly developed evolutionary trait.

Our screen of RNA segments was not wide enough to conclude which RNA segments are required for amyloid formation, as it was not the focus of this work. Nevertheless, we were able to provide some suggestions in the 2nd and 4th discussion paragraphs, based on our observations. They are as follows:

Revised text in the Discussion (pages 9-10 lines 252-260, 267-268 & 274-280):

“The central LCD of NCAP forms fibrils with the long and structured S2hp vRNA segment (Figure 1; Table S1), with various short, single-stranded RNA sequences (Figure 2), and also with no RNA (Figure 1. f). This suggests that specific LCD-RNA interactions are not required for LCD-amyloid formation. Nevertheless, the LCD does bind to at least S2hp vRNA¹, and LCD fibril maturation is influenced by the RNA sequence and length (Figure 2. b), so LCD-RNA interactions play a role. The LCD segment is highly positively charged (Figure 1. b), especially in its non-phosphorylated form. Therefore we expect it to engage in non-specific polar interactions with the negatively charged RNA, which in turn may promote the accumulation of LCD molecules, including through PS formation (Figure 3), and their amyloid-like assembly (Figure 2. a).

...

Full-length NCAP is capable of only scarce fibril formation in the presence and absence of S2hp vRNA and with ZnCl₂ (Figure S2; Table S1). ... In a parallel study, we show that the structured regions of S2hp are essential for strong binding to NCAP, whereas S2 and other short, single-stranded RNA segments bind to it weakly¹. Indeed, with the short, unstructured vRNA segments S1, S1.5 and S2 (Table S1) we did not detect fibrils of full-length NCAP or increased ThT signal in preliminary experiments (data not shown), although those showed fibril formation of the LCD segment (Figure 2. b). We therefore speculate that robust amyloid formation of NCAP requires strong interactions with specific vRNA sequences and/or co-factors that we are yet to identify.”

2. The ability of the full NCAP protein to form amyloid fibrils is not as convincingly proven as it is with the LCD domains.

True. This is our repeated finding. We state this in the Results (page 5 lines 127-129) as well as in the Discussion (see above in the answer to point #1; pages 9-10 lines 252-260, 267-268 & 274-280 in the revised ms) and moved our findings on full NCAP to the Supplement so to focus the manuscript on the robust ability of the central LCD to form amyloid. This focus on the central LCD is also reflected in the revised Abstract and Title. Considering that we have some indications of fibril formation with the full-length protein, especially in low ionic strength buffer (Figure S2), as well as some reaction with the amyloid dyes ThT and ThS, and given the robust ability of the LCD segment to form fibrils, we believe that we have not yet identified the appropriate vRNA segment or co-factors for a more robust NCAP fibril formation.

Results section (page 5 lines 127-129):

“...NCAP and also the DD-C_{term} fibrils are much scarcer in EM images compared to the central LCD, and their morphologies differ from those of the central LCD or canonical amyloid fibrils.”

Abstract:

“The self-assembly of the Nucleocapsid protein (NCAP) of SARS-CoV-2 is crucial for its function. Computational analysis of the amino acid sequence of NCAP reveals low-complexity

domains (LCDs) akin to LCDs in other proteins known to self-assemble as phase separation droplets and amyloid fibrils. Previous reports have described NCAP's propensity to phase-separate. Here we show that the central LCD of NCAP is capable of both, phase separation and amyloid formation. Within this central LCD we identified three adhesive segments and determined the atomic structure of the fibrils formed by each. Those structures guided the design of G12, a peptide that interferes with the self-assembly of NCAP and demonstrates antiviral activity in SARS-CoV-2 infected cells. Our work, therefore, unveils the amyloid form of the central LCD of NCAP and suggests that amyloidogenic segments of NCAP could be targeted for drug development."

Title:

"LOW COMPLEXITY DOMAINS OF THE NUCLEOCAPSID PROTEIN OF SARS-COV-2 FORM AMYLOID FIBRILS"

a. The EM images of the fibrils in NCAP figure 1D are unconvincing (especially the +vRNA), I would suggest these are replaced with better images that more clearly show the presence of amyloid-like fibrils.

Unlike fibrils of the central LCD, the fibrils of the full length NCAP are rare and non-amyloid typical, as we state in the revised text (see above in the answer to point #2; page 5 lines 127-129 in the revised ms). In repeating experiments we saw similar fibrils of NCAP scarcely found on the EM grid, apart for the denser fibrils we saw in weak ionic strength buffer in the presence of ZnCl₂ (Figure S2. d). We speculate that robust amyloid formation of NCAP is possible, but we are yet to identify the conditions and co-factors needed for it. This speculation is based on the clear ability of the central LCD of NCAP to form amyloid-like fibrils, similarly to fibrils of LCDs of other amyloid-forming RNA-binding proteins²⁻⁷, and on observations of some fibrils and amyloid-dye binding by the full-length NCAP. We now discuss those limitations in the paper (page 5 lines 127-129 & pages 9-10 lines 252-260, 267-268 & 274-280), and moved the results about the full length NCAP to the Supplementary Information.

b. In figure 1c the NCAP +vRNA curve looks strange for a relatively large protein due to the lack of lag phase. To me it looks much more like the aggregation of a small amyloidogenic peptide. The authors should provide an explanation as to why the ThT curve of NCAP + vRNA looks so different to LCD + vRNA. Also why do none of the proteins or fragments show +ve ThT signals in the absence of vRNA, the EM images of LCD -vRNA seem to show a high abundance of fibrils, you would expect this to result in some ThT signal?

In the revised Results section (page 4 lines 108-112), we note the lack of a lag phase, as follows:

"However, neither NCAP nor the DD-C_{term} segment demonstrated a clear lag phase in their ThT curves. Also, in the absence of S2hp vRNA, we did not detect an increase in ThT fluorescence

in any of the samples within the 33 h of measurements. This suggests that vRNA promotes the formation of ThT positive aggregates from those LCD containing protein constructs, at least in the first 1.5 days of incubation.”

We also comment in the Discussion (pages 9-10, lines 269-274):

“..., the NCAP+ S2hp aggregates produce a ThT amyloid formation curve, but it lacks a lag phase (Figure S2. e). Short or absent lag phase in ThT curves may result from the existence of pre-formed amyloid seeds in the tested sample⁸, or from a fast pickup of the ThT signal prior to starting the measurements. The latter may be reasonable given that NCAP rapidly aggregates and becomes turbid in the presence of S2hp (Figure S2. f). “

The fibrils that formed with no RNA formed at higher protein concentration and longer incubation times compare to the conditions for the ThT experiment. In the revised text we describe this in Results (pages 4-5, lines 114-123):

“To observe fibrils of NCAP and its LCD containing segments by EM we increased protein concentration and incubated each protein separately for ~1- 2 weeks with and without S2hp vRNA. Of note, under the conditions used for the ThT experiment we did not detect fibrils by EM, suggesting that the ThT experiment is more sensitive for the detection of amyloid-like aggregates or that ThT interacts with pre-fibrillar assemblies of the proteins. Other explanations, such as poor adherence of the protein fibrils to the EM grid and fibril reversibility are also reasonable. Nevertheless, fibrils were detected by EM both in the presence and absence of the vRNA, unlike the positive ThT signal which was only observed in the presence of the vRNA. This indicates that vRNA is not essential for fibril formation, but may accelerate it. “

c. In figure 3 fiber diffraction is only shown for the LCD segment, were the authors able to detect the 10 and 4.7 Å reflections from the full NCAP protein? If so this should be shown, if not some explanation should be provided.

We were unable to obtain a clear diffraction pattern for NCAP as we state in the revised Results section (page 5, lines 140-143). We therefore focused the paper more on the LCD segment itself for which we proved amyloid formation, as described above in our answer to point #2.

The revised paragraph in Results reads:

“... Nevertheless, we were unable to obtain a clear diffraction pattern from the full-length NCAP. This may be a result of low fibril concentration, as evident by EM (Figures 1. f & S2. c), and/or from fibril decomposition during washing steps meant to eliminate salts from the sample.”

3. I believe this improved characterization of NCAPs fibrillation propensity is important as the antiviral activity experiments in figure 5 are based on the full NCAP protein. Clearer images of

fibril formation in Figure 5 would make it easier to argue that the G12 peptide is disrupting NCAP's ability to form amyloids, which is then going on to prevent PS (the DIC images very clearly show this) and reduce viral infection.

We decided to omit those results from the paper, the reason is that G12 is dissolved in DMSO and the addition of DMSO to NCAP allowed the formation of the fibril clusters we presented in the original manuscript. We were unable to obtain clearer EM images of NCAP fibrils with G12. G12 was designed to interact with the central LCD based on its tendency to form steric-zipper structures, and thereby interfere with NCAP's self-assembly. The LCD segment may also contribute to NCAP's PS formation (Figure 3. a and S3). PS of NCAP was shown to occur in NCAP-transfected and SARS-CoV-2 infected cells⁹⁻¹², and addition of G12 disrupts the ordered PS formation of NCAP (Figure 5. b). We therefore believe that the NCAP-PS disrupting ability of G12 damages the function of the protein within infected cells which leads to reduced viral infection in the culture.

We emphasize this point in the revised Result section (page 8, lines 218-220):

“To modulate NCAP's self-assembly we exploited the propensity of NCAP's LCD to form steric-zipper structures. Guided by our amyloid-spine structures we screened an array of peptides, each designed to interact with a specific steric-zipper forming segment....”

And in the Discussion (page 11, line 305-308):

“By targeting the amyloidogenic segment ₂₁₇AALALL₂₂₂ (Figures 4 & S4-S5) with G12, we inhibited the PS formation of NCAP in-vitro (Figure 5. b-c). G12 is a peptide designed to interact and block the ₂₁₇AALALL₂₂₂ interface by exploiting the tendency of this segment to form steric-zipper structures (Figure 5. a).”

Reviewer #2 (Remarks to the Author):

The manuscript “Low Complexity Domains of the Nucleocapsid Protein of SARS-CoV-2 Drive Amyloid Formation” by Fligelman...Eisenberg describes a structure-function analysis of the SARS-CoV-2 nucleocapsid (NCAP) protein, in particular identifying and characterizing the proteins' propensity to form amyloid fibrils. NCAP is highly abundant in SARS-CoV-2 cells, is highly immunogenic, and is known to play crucial roles in both the production of viral RNAs and their packaging into virions. Because of its importance to different life cycle stages and its immunogenicity, NCAP has recently gained more attention as a possible target for next-generation SARS-CoV-2 vaccines and antiviral therapies.

In this paper, Fligelman et al. follow up a number of studies that have demonstrated that NCAP forms phase separated condensates both on its own and with viral (and other) RNAs. They show

that the central unstructured domain (termed LCD here) is principally responsible for this behavior, and critically this is the first report to show that this region promotes the formation of amyloid fibrils in addition to less-specific condensates demonstrated by others. Drawing on their expertise in this area, they determine high-resolution 3D structures of three short regions in the LCD that form fibrils. Based on these structures, they design a peptide inhibitor of NCAP self-assembly and demonstrate that it has measurable antiviral activity in a cell culture model of SARS-CoV-2 infection.

Overall, this paper is important and timely, the data and presentation are solid and clear, and I recommend publication with only minor changes, detailed below.

There are several points in the manuscript which would benefit from more explanation. 1. One such point is the computational analysis of NCAP described in the first sentence of the Results section. How was this analysis done? What features is the analysis looking for? How successful is this type of analysis in accurately identifying regions that undergo phase separation/amyloid formation?

We have revised the first paragraph of the Results section (page 3, lines 78- 86) to address those questions, as follows:

“Using the SEG algorithm¹³ we analyzed the sequence of NCAP and identified a 75-residue LCD (residues 175-249) within NCAP’s central intrinsically disordered region, as well as a second, lysine-rich LCD of 19 residues (residues 361-379) within its C-terminal tail (CTT) (Figure 1. a-b). SEG is a widely used algorithm that identifies segments in a sliding window as either high or low complexity by statistically analyzing the amino acid distribution as a measure of sequence complexity¹³. While not all LCDs identified this way are capable of PS and amyloid formation, LCDs which do phase separate are readily identified by SEG^{14,15}. In NCAP, those central and C-terminal LCDs, along with an N-terminal disordered region, flank the structured RNA-binding and dimerization domains of the protein (Figure 1. a).”

2. Next, the manuscript would benefit from more explanation of the choice of viral RNAs used in the study. Late in the manuscript, the authors refer to a related manuscript in bioRxiv (which is itself very interesting) that details the dissection of the 5’ end of the viral RNA and its interactions with NCAP. A few sentences describing this work and how it affected the choice of RNAs for this study would help the reader.

We have added an explanation to our choice of RNA in the Results section (page 4, line 94- 99):

“We then verified that our purified full length NCAP protein is capable of PS by mixing it with a 211-nucleotide 5’- genomic vRNA segment named hairpin-Site2 (S2hp; Figure S1, Table S1) in the presence and absence of the PS enhancing $ZnCl_2$ ¹⁶ (Figure S2. a and Supplementary text). The

S2 vRNA sequence was previously suggested to be a strong NCAP cross-linking site¹¹, and we extended it by including the adjacent hairpin regions that improve binding to NCAP¹.”

3. *Figure 1A: What do the capital letters (RM..ESKMS) represent in the sequence of the LCD? Are these residues that are mutated in different variants? Also on this figure panel, the authors may consider labeling/underlining the six-residue segments they identify as contributing to amyloid fibril formation.*

Thank you for this comment. We edited the figure and figure legend accordingly (page 14, lines 338- 391):

“Lowercase letters represent residues of low-complexity while capital letters represent non-low-complexity residues. No more than 5 interrupting non-low-complexity residues between strings of 10 or more low-complexity residues were allowed. Steric-zipper forming sequences that are discussed below are underlined in the sequence of the central LCD.”

4. *In the discussion, the authors should address the question of NCAP phosphorylation: where are the identified phosphorylation sites relative to the identified amyloid-forming segments? How do the authors envision phosphorylation to alter the self-assembly properties of NCAP?*

We address these questions in the Discussion (page 10, lines 284-293) as follows:

“...The segment ¹⁷⁹GSQASS₁₈₄ is part of a conserved serine/arginine (SR) rich region (residues 176-206)¹⁰ and it includes the two phosphorylation sites S180 and S184¹². Phosphorylation of the SR-rich region facilitates the transformation of NCAP’s PS droplets from solid to liquid-like states during viral genome processing. The non-phosphorylated protein, however, is associated with solid PS droplets and nucleocapsid assembly¹⁷. Both S180 and S184 face the dry, tight interface formed between the β-sheets in the structure of ¹⁷⁹GSQASS₁₈₄ (Figure 4. a). Phosphorylation of those residues is indeed likely to reverse the solid, amyloid-like packing of this segment. Of note, all results in this paper showing the ordered, solid-like mode of aggregation were obtained with non-phosphorylated proteins and peptides. “

5. *In a related point, there are a huge number of SARS-CoV-2 genome sequences available, and those of a number of related coronaviruses as well. How well-conserved are the identified self-associating segments within SARS-CoV-2 isolates? Are any of these regions mutated in more recently-evolved variants? Related to this, are there regions of related coronavirus N proteins that are predicted to drive similar self-assembly? This would be expected if the amyloid fiber formation were to directly contribute to viral RNA packaging.*

To address these questions, we have added Figure S9 to the Supplement and two paragraphs to the Discussion (pages 11-12, lines 318- 344).

“The three steric-zipper-forming segments we identified in this work are conserved between the NCAPs of SARS-CoV-2 and SARS-CoV. The only exception is alanine in position 217 in the sequence of SARS-CoV-2 which is replaced by threonine in the NCAP of SARS-CoV (Figure S9. a). A ZipperDB calculation on the LCD of the NCAP of SARS-CoV revealed that this threonine shifts the steric-zipper forming segment to the hydrophobic ALALL sequence (with Rosetta free energy score of -24.700) that is aligned and conserved with residues 218 - 223 in the NCAP of SARS-CoV-2. This suggests that the LCD in the NCAP of SARS-CoV may also form amyloids, and that future SARS coronaviruses might share this targetable property. A SEG analysis¹³ performed on the sequence of the NCAPs of a number of α , β and γ coronaviruses from various species showed that many of these viruses contain LCDs that could potentially participate in amyloid formation (Figure S9. b). This suggests that amyloid formation of NCAP LCDs is a general mechanism of action and a common targetable trait in coronaviruses.

Despite the high conservation of NCAP¹⁸, some mutations have been identified in strains that emerged since the initial SARS-CoV-2 outbreak in Wuhan, China. To date, no NCAP mutations were detected within our amyloid steric-zipper spine segments: ₁₇₉GSQASS₁₈₄, ₂₁₇AALALL₂₂₂, and ₂₄₃GQTVTK₂₄₈. Nevertheless, some mutations were detected within the central LCD, including the prevalent R203K/M, G204R/M and T205I substitutions¹⁹⁻²¹. The R203K/G204R mutants exhibit higher PS propensity compared to the Wuhan variant⁹, and the R204M mutation promotes RNA packaging and viral replication in the delta variant²². Also interesting are the G214C (Lambda variant) and G215C (Delta variant) substitutions¹⁹⁻²¹ that are adjacent to the ₂₁₇AALALL₂₂₂ steric-zipper segment. The Delta variant spread faster and caused more infection compared to its predecessors²³⁻²⁶. The Delta variant also carries a D377Y mutation in the C-terminal LCD segment of NCAP. It is possible that mutations in NCAP’s LCD enhance amyloid formation, similarly to mutations in other RNA-binding proteins^{2,27-29}. This is important to explore since amyloid fibrils are associated with numerous dementias and movement disorders^{30,31}. Amyloid cross-talk and hetero-amyloid aggregation, including between microbial and human amyloid proteins (e.g.³²⁻³⁵), is a well-known phenomenon that is postulated to exacerbate amyloid pathology³⁶. “

Reviewer #3 (Remarks to the Author):

The manuscript by Tayeb-Fligelman et al. reported that the Nucleocapsid protein (NCAP) of SARS-CoV-2 can form amyloid fibrils, and inhibiting amyloid formation of NCAP by the designed

peptide can diminish SARS-CoV-2 infection. Amyloid fibrillation of different proteins has been observed in different human tissues (e.g., brain, heart and skin), which is closely related to various human diseases. In this study, the authors found that NCAP of SARS-COV-2 also exhibits the capability of forming amyloid fibrils, which is an important finding. They further demonstrated that the central low-complexity domains (LCD) of NCAP can undergo liquid-like phase separation to form Thioflavin-S positive droplets and then transit to solid-like amyloid fibrils which exhibit typical intermolecular β -sheet packing features. Furthermore, the authors determined the atomic structures of three adhesive segments in the central LCD. Importantly, the authors designed a peptide which can interfere with the self-assembly of NCAP and exhibits dose-dependent antiviral activity in the cell model, which confirms the potential role of NCAP fibrillation in virus infection and on the other hand suggests a new antiviral strategy of targeting NCAP fibrillation. Overall, this work revealed the important role of the LCD of NCAP in driving its phase separation and liquid-to-solid phase transition, and indicated the potential of inhibiting the fibrillation process of NCAP for preventing virus infection. Whereas, it remains unclear whether the fibrillar form of NCAP indeed exists during virus infection and replication. To strengthen this work, the authors need to address the concerns listed below:

Major concerns:

1. Due to the poor quality and low resolution of the NS-TEM images (e.g., Nterm-RBD+vRNA in Figure 1D; NCAP+S2hp in Figure S2B iv.), it's hard to clearly see the details of the sample, and it's insufficient to distinguish whether these samples formed amorphous aggregates or amyloid fibrils. The authors need to provide NS-TEM images with high quality.

We checked the fibril formation of NCAP numerous times and observed similar fibril morphologies. Fibrils formed in the presence of S2hp were usually coated with either aggregated protein or the RNA itself while some exposed regions and fibrils convinced us of NCAP fibril formation. We could not obtain better images. We therefore decided to move the results with the full-length NCAP to the Supplement and clearly acknowledge the limitation in the text (please see answer to Reviewer #1 points 2& 3). We also omitted results with N_{term}-RBD and are now focusing the manuscript on the unambiguous ability of the central LCD to form amyloid fibrils *in vitro*.

We revised the Results section accordingly (pages 4-5, lines 87-131):

“NCAP’s LCDs PARTICIPATE IN FIBRIL FORMATION

To assess possible amyloid formation of NCAP’s LCDs and to identify adhesive segments that drive it, we expressed and purified NCAP and its LCD containing segments in *E. coli*. Those segments included residues comprising NCAP’s central LCD and surrounding residues (construct named LCD, residues 171-263) and a segment that includes the C-terminal LCD with the C-

terminal tail and dimerization domain (construct DD-C_{term}, residues 257-419) (Figure 1. c). Only RNA-free protein fractions were combined at the last step of protein purification for use in subsequent experiments. We then verified that our purified full length NCAP protein is capable of PS by mixing it with a 211-nucleotide 5'- genomic vRNA segment named hairpin-Site2 (S2hp; Figure S1, Table S1) in the presence and absence of the PS enhancing ZnCl₂¹⁶ (Figure S2. a and Supplementary text). The S2 vRNA sequence was previously suggested to be a strong NCAP cross-linking site¹¹, and we extended it by including the adjacent hairpin regions that improve binding to NCAP¹.

Using our recombinant protein system we found that NCAP's LCDs are capable of binding the amyloid-dye Thioflavin-T (ThT). In a ThT amyloid-formation kinetic assay performed over ~33 h of measurement (Figure 1. d-e), S2hp vRNA mixtures (in 4:1 protein: vRNA molar ratio) of the central LCD and the DD-C_{term} segments of NCAP produced amyloid formation curves. Whereas the DD-C_{term}+ vRNA curve plateau after ~ 3 h of incubation, LCD+ vRNA plateaus ~ 10 h after the start of measurements while producing a significantly higher fluorescence signal than that of DD-C_{term}. The full length NCAP also exhibited increased ThT fluorescence over 5 h of measurement when mixed with S2hp vRNA, followed by a slight decrease in signal, possibly because of spontaneous disaggregation (Figure S2. e). However, neither NCAP nor the DD-C_{term} segment demonstrated a clear lag phase in their ThT curves. Also, in the absence of S2hp vRNA, we did not detect an increase in ThT fluorescence in any of the samples within the 33 h of measurements. This suggests that vRNA promotes the formation of ThT positive aggregates from those LCD containing protein constructs, at least in the first 1.5 days of incubation.

Visualization of fibrils by electron microscopy (EM) confirmed the propensity of the LCD-containing constructs to adopt fibrillar morphologies (Figures 1. f & S2. c). To observe fibrils of NCAP and its LCD containing segments by EM we increased protein concentration and incubated each protein separately for ~1- 2 weeks with and without S2hp vRNA. Of note, under the conditions used for the ThT experiment we did not detect fibrils by EM, suggesting that the ThT experiment is more sensitive for the detection of amyloid-like aggregates or that ThT interacts with pre-fibrillar assemblies of the proteins. Other explanations, such as poor adherence of the protein fibrils to the EM grid and fibril reversibility are also reasonable. Nevertheless, fibrils were detected by EM both in the presence and absence of the vRNA, unlike the positive ThT signal which was

only observed in the presence of the vRNA. This indicates that vRNA is not essential for fibril formation, but may accelerate it.

The full-length NCAP also formed fibrillar morphologies in samples containing higher protein to vRNA ratio (40: 1 protein: vRNA molar ratio), as well as when incubated with zinc ions in PBS (Figure S2. b), and particularly in a low ionic strength buffer (Figure S2. d) upon 3- 6 days of incubation (as indicated in Figure S2). NCAP and also the DD-C_{term} fibrils are much scarcer in EM images compared to the central LCD, and their morphologies differ from those of the central LCD or canonical amyloid fibrils. Together, those observations suggest that NCAP and its LCD containing segments, particularly the central LCD segment, are capable of forming aggregates of fibrillar morphologies as well as ThT positive species.”

2. Figure 1D presents NS-TEM images of samples after incubation of 2 weeks to support the accelerating activity of vRNA in NCAP amyloid fibril formation. While, Figure 1C shows the results of ThT fibril formation kinetic assay within 33 hours, and the ThT curve reaches the peak within 6-8 hour and keeps decreasing after then. The authors need to address this. Whether NCAP forms fibrils when ThT intensity peaks. Moreover, from the TEM images (Fig.1D), the NCAP+vRNA sample doesn't form typical amyloid fibrils, they look more like oligomers. Thus, the authors may need to provide additional evidence to validate that NCAP indeed forms typical amyloid fibrils rather than certain types of oligomers or precipitates.

Please see our answer to your point 1. In addition, in the revised paper (pages 4-5, lines 114-120) we now made it clear that we used higher protein concentrations and incubation times to obtain the EM images shown in Figure 1 and S2. c and suggest an explanation:

“To observe fibrils of NCAP and its LCD containing segments by EM we increased protein concentration and incubated each protein separately for ~1- 2 weeks with and without S2hp vRNA. Of note, under the conditions used for the ThT experiment we did not detect fibrils by EM, suggesting that the ThT experiment is more sensitive for the detection of amyloid-like aggregates or that ThT interacts with pre-fibrillar assemblies of the proteins. Other explanations, such as poor adherence of the protein fibrils to the EM grid and fibril reversibility are also reasonable. “

3. It's interesting that the authors report the phase separation and transition process of LCD in the presence of different molar ratio of S2hp by using different characterizing methods (Fig.2). To ensure that LCD eventually condenses into amyloid fibrils from droplet, the authors need to provide TEM images for each time point, especially the end point, where LCD droplets mostly disappear and ThS-positive puncta massively forms.

As mentioned in our response to point 2 of Reviewer #3, we could not detect fibrils in EM at this protein concentration and incubation time without RNA or in the presence of S2hp. Given this limitation we do not specifically claim that LCD fibrils are grown within PS droplets, only that the LCD is capable of both liquid to solid PS formation and amyloid formation. In the revised paper we clearly state in the Results (page 7, lines 180-181) our inability to see fibrils by EM using samples shown in the current Figure 3. a, as follows:

“Here too, no fibrils could be detected by EM at the concentration and incubation times used for the PS assay.”

And summarize those results in a manner that refers to the properties of the PS particles rather than to the structure or nature of the aggregates consisting those particles (page 7, lines 190-193):

“Overall, our results indicate that the central LCD of NCAP forms ThS-positive PS droplets that transition from circular liquid droplets to fibrous or amorphous solid-like particles, and that the RNA concentration governs the kinetics of this process and the morphology of the assemblies.”

4. The authors demonstrated the roles of viral RNA in accelerating the fibril formation process of central LCD and NCAP. While, the mechanism of vRNA recognition, binding to NCAP and promoting the self-assembly process remains unclear. The authors may need to discuss this point to strengthen the interesting finding of this work.

In the revised paper, we included Discussion paragraphs (pages 9-10, lines 252- 280) that address the interactions of vRNA with NCAP and its LCD in relation to fibril formation (see below). We note that because of safety regulations we could not work with the entire genomic RNA and it is very possible that different RNA segments will bind better to NCAP and possibly result in more fibril formation.

“The central LCD of NCAP forms fibrils with the long and structured S2hp vRNA segment (Figure 1; Table S1), with various short, single-stranded RNA sequences (Figure 2), and also with no RNA (Figure 1. f). This suggests that specific LCD-RNA interactions are not required for LCD-amyloid formation. Nevertheless, the LCD does bind to at least S2hp vRNA¹, and LCD fibril maturation is influenced by the RNA sequence and length (Figure 2. b), so LCD-RNA interactions play a role. The LCD segment is highly positively charged (Figure 1. b), especially in its non-phosphorylated form. Therefore we expect it to engage in non-specific polar interactions with the negatively charged RNA, which in turn may promote the accumulation of LCD molecules, including through PS formation (Figure 3), and their amyloid-like assembly (Figure 2. a).

The amyloid-like characteristics of the central LCD of NCAP are similar to those of the LCDs of FUS^{2,3}, hnRNPA2⁴, TDP-43⁵, and other RNA-binding proteins that are involved in RNA metabolism in eukaryotic cells^{6,7}, and under certain circumstances, also in amyloid-associated pathologies³⁷. This equivalent ability of the LCD of NCAP to PS and stack into amyloid-like

structures in the presence of RNA proposes its potential function in the yet elusive mechanism of NCAP self-assembly.

Full-length NCAP is capable of only scarce fibril formation in the presence and absence of S2hp vRNA and with ZnCl₂ (Figure S2; Table S1). Whereas fibrils formed in the presence of S2hp do not exhibit amyloid-typical morphology (Figure S2. b-c), the NCAP+ S2hp aggregates produce a ThT amyloid formation curve, but it lacks a lag phase (Figure S2. e). Short or absent lag phase in ThT curves may result from the existence of pre-formed amyloid seeds in the tested sample⁸, or from a fast pickup of the ThT signal prior to starting the measurements. The latter may be reasonable given that NCAP rapidly aggregates and becomes turbid in the presence of S2hp (Figure S2. f). In a parallel study, we show that the structured regions of S2hp are essential for strong binding to NCAP, whereas S2 and other short, single-stranded RNA segments bind to it weakly¹. Indeed, with the short, unstructured vRNA segments S1, S1.5 and S2 (Table S1) we did not detect fibrils of full-length NCAP or increased ThT signal in preliminary experiments (data not shown), although those showed fibril formation of the LCD segment (Figure 2. b). We therefore speculate that robust amyloid formation of NCAP requires strong interactions with specific vRNA sequences and/or co-factors that we are yet to identify.”

5. Increasing concentrations of G12 were found to disrupt the phase separation process but promote the formation of larger aggregates as shown in Figure 5. This raises a question: are these larger aggregates ThS-positive amyloid aggregates? The authors may need to address this by carefully characterizing the nature of these aggregates.

Aggregates formed in the presence of G12 showed lower ThS signal (see image below, not included in the paper). Nevertheless, following the comment of Reviewer #1’s point 3 regarding the fibrils of NCAP with G12, we decided to omit those results from the paper and focus on G12’s interference with the PS of NCAP. PS of NCAP is well established, including in NCAP transfected and SARS-CoV-2 infected cells⁹⁻¹². G12, designed to interact with the steric-zipper forming segment AALALL of the LCD, can abrogate this PS formation.

Figure legend: G12 disrupts NCAP's phase separation and ThS reactivity. NCAP protein was mixed with S2hp RNA and ZnCl₂. ThS was added at a final concentration of 0.0002 % w/v. G12 was added to the appropriate mixture in 1:1 NCAP: inhibitor molar ratio. Droplets were imaged after 30 min of incubation. Top: LLPS droplets of NCAP with no inhibitors show substantial ThS partitioning (green). Bottom: addition of G12 completely disrupts NCAP's LLPS and induces formation of protein aggregates with reduced ThS binding capacity.

6. *Lower concentrations of G12 (0.5µM, 1.5µM, 4µM) shows an increasing relative infected cell rate (>100%) in Figure S7 and this data needs to be explained further.*

In the revised paper, we acknowledged the increased viral infectivity in low G12 concentrations in the Result section (Page 8 lines, 238- 241):

“Whereas G12 concentrations lower than 6 µM slightly increase the relative percent infectivity of treated cells, in the range of ~ 6-16 µM, G12 exhibits dose-dependent antiviral activity while reducing the amount of virus detected in the culture by up to ~ 50% without inflicting cytotoxicity (Figures 5. d & S8).”

And have added to the Discussion (page 11, lines 311-317) our speculation for it:

“G12 concentrations lower than 6 µM, however, led to increased viral infection in treated cells. We speculate that when administered in subeffective concentrations, G12 partitions into NCAP droplets and increases NCAP's effective concentration which possibly promotes self-assembly and formation of new virions. When administered in proper concentrations, we anticipate that the antiviral activity of G12 results from its interference with the self-assembly of NCAP, as designed, leading to poor RNA-packaging and viral particle-assembly.”

7. *To what extent do fibrils from the central LCD resemble those formed by NCAP with vRNA? The author may need to discuss.*

Using fibril diffraction methods, we proved that the LCD fibrils are typical amyloid fibrils. Under our experimental conditions, only ThT and ThS binding suggest the amyloid properties of full length NCAP. We clearly state this in the revised paper and moved results with NCAP to the Supplementary Information so as to focus on the amyloid formation of the LCD and its use in the design of NCAP manipulators of self-assembly.

8. *The incubation time of NCAP with and without ZnCl₂ in Figure S2C was not provided.*

We thank the reviewer for noticing that. This information was available before in the method section, and is now also in the figure legend (now Figure S2. d; 3 days of incubation).

Minor concerns:

1. In the *RESULTS* section, the word “PS” on line 121 used an abbreviation but the full name “PHASE SEPARATION” was used on line 204.

We thank the reviewer for pointing this out. We found a few more lines with the same issue and corrected it throughout.

2. The word “DROPLES” on line 121 should be “DROPLETS”.

Thank you for noticing this typo. We corrected it.

Summary of formatting changes and general edits:

1. Author affiliations now refer to only one address/department and organized according to their appearance in the author list.
2. Present addresses are now supplied below the list of affiliations.
3. The abstract was contracted to consist less than 150 words and edited to reflect the changes we made to the manuscript.
4. There are changes in references numbers compared to the previous version. Those changes are not highlighted in green.
5. The order of the last two introductory paragraphs (page 3) was reversed. The introduction now ends with the paragraph describing our main findings.
6. Titles of all result sections were contracted to consist 60 characters or less.
7. The text under the result section “NCAP’s LCDs PARTICIPATE IN FIBRIL FORMATION” was edited extensively since the results with the full-length NCAP were moved to the Supplement, and results with the construct N_{term} -RBD were omitted from the revised manuscript following reviewers’ comments.
8. The order of the paragraphs under the result section “THE CENTRAL LCD FORMS AMYLOID TYPICAL FIBRILS” (page 5) was reversed for better readability of the revised paper.
9. The order of the result sections: “NCAP’S CENTRAL LCD FORMS PS DROPLETS AND SOLID PARTICLES“ and “THE CENTRAL LCD FORMS AMYLOID TYPICAL FIBRILS” was reversed. Figure numbers were updated accordingly.
10. We reorganized and revised the Discussion so as to better focus on the amyloid formation of the central LCD of NCAP and to address specific reviewers’ concerns. Edits made following reviewers’ comments are detailed above.

11. Panel labeling in all figures was changed to the a, b, c convention and figure legends were revised accordingly.
12. Figure 3 of the revised paper- bar graphs are now showing information about the distribution of the underlying data (Figure 3. c), or were replaced by a box-and-whisker plot (Figure 3. b).
13. Information regarding statistical parameters (n, t, df) are now provided in the legend of appropriate figures (Figures 1, 3 & 5).
14. The method section was edited according to the revisions made to the paper, including the addition of sequence conservation analysis of the LCDs of the NCAPs of SARS-CoV and SARS-CoV-2 (Figure S9. a), identification of LCDs in other coronaviruses (Figure S9. b), and deletion of methods related to data removed from the paper (the effect of G12 on fibril formation of NCAP and results with the segment N_{term}-RBD).
15. Subheadings in the method sections were contracted to include 60 characters or less.
16. The Supplementary Results and Discussion section in the Supplement was edited to explain results with the full length NCAP that were moved to the Supplement (Figure S2) and to delete discussion regarding the N_{term}-RBD segment that was omitted from the revised paper.
17. Figure S2 in the Supplement was edited to include data that was previously shown in the main text (panels c and e of the new Figure S2). The Figure legend was edited accordingly.
18. A stereo view of atomic structures of the steric-zipper forming segments of the central LCD was added to the Supplement (Figure S5).
19. A figure showing the sequence of a fragment of the constructs that was used to express SUMO_tagged, full length NCAP protein was added to the Supplement (Figure S10).
20. A figure showing aligned translations of Sanger sequencing reads that fully cover fragments of the NCAP gene was added to the Supplement (Figure S11).
21. Data availability: we added links to the RCSB- Protein Data Bank entries of our steric zipper structures.
22. Funding sources information was added to the Acknowledgment section.

References:

1. Tai, C. E., Tayeb-Fligelman, E., Griner, S., Salwinski, L., Bowler, J. T., Abskharon, R., Cheng, X., Seidler, P. M., Jiang, Y. X., Eisenberg, D. S. & Guo, F. The SARS-CoV-2 nucleocapsid protein preferentially binds long and structured RNAs. *bioRxiv* doi:10.1101/2021.12.25.474155 (2021).
2. Luo, F., Gui, X., Zhou, H., Gu, J., Li, Y., Liu, X., Zhao, M., Li, D., Li, X. & Liu, C. Atomic structures of FUS LC domain segments reveal bases for reversible amyloid fibril formation. *Nat. Struct. Mol. Biol.* **25**, 341–346 (2018).
3. Murray, D. T., Kato, M., Lin, Y., Thurber, K. R., Hung, I., McKnight, S. L. & Tycko, R. Structure of FUS Protein Fibrils and Its Relevance to Self-Assembly and Phase Separation

- of Low-Complexity Domains. *Cell* **171**, 615–627.e16 (2017).
4. Lu, J., Cao, Q., Hughes, M. P., Sawaya, M. R., Boyer, D. R., Cascio, D. & Eisenberg, D. S. CryoEM structure of the low-complexity domain of hnRNPA2 and its conversion to pathogenic amyloid. *Nat. Commun.* **11**, 4090 (2020).
 5. Guenther, E. L., Cao, Q., Trinh, H., Lu, J., Sawaya, M. R., Cascio, D., Boyer, D. R., Rodriguez, J. A., Hughes, M. P. & Eisenberg, D. S. Atomic structures of TDP-43 LCD segments and insights into reversible or pathogenic aggregation. *Nat. Struct. Mol. Biol.* **25**, 463–471 (2018).
 6. Kato, M., Han, T. W., Xie, S., Shi, K., Du, X., Wu, L. C., Mirzaei, H., Goldsmith, E. J., Longgood, J., Pei, J., Grishin, N. V., Frantz, D. E., Schneider, J. W., Chen, S., Li, L., Sawaya, M. R., Eisenberg, D., Tycko, R. & McKnight, S. L. Cell-free Formation of RNA Granules: Low Complexity Sequence Domains Form Dynamic Fibers within Hydrogels. *Cell* **149**, 753–767 (2012).
 7. Hughes, M. P., Sawaya, M. R., Boyer, D. R., Goldschmidt, L., Rodriguez, J. A., Cascio, D., Chong, L., Gonen, T. & Eisenberg, D. S. Atomic structures of low-complexity protein segments reveal kinked β sheets that assemble networks. *Science* **359**, 698–701 (2018).
 8. Holubová, M., Štěpánek, P. & Hrubý, M. Polymer materials as promoters/inhibitors of amyloid fibril formation. *Colloid Polym. Sci.* **299**, 343–362 (2021).
 9. Zhao, M., Yu, Y., Sun, L.-M., Xing, J.-Q., Li, T., Zhu, Y., Wang, M., Yu, Y., Xue, W., Xia, T., Cai, H., Han, Q.-Y., Yin, X., Li, W.-H., Li, A.-L., Cui, J., Yuan, Z., Zhang, R., Zhou, T., *et al.* GCG inhibits SARS-CoV-2 replication by disrupting the liquid phase condensation of its nucleocapsid protein. *Nat. Commun.* **2021** **12**, 1–14 (2021).
 10. Lu, S., Ye, Q., Singh, D., Cao, Y., Diedrich, J. K., Yates, J. R., Villa, E., Cleveland, D. W. & Corbett, K. D. The SARS-CoV-2 nucleocapsid phosphoprotein forms mutually exclusive condensates with RNA and the membrane-associated M protein. *Nat. Commun.* **12**, 502 (2021).
 11. Iserman, C., Roden, C. A., Boerneke, M. A., Sealfon, R. S. G., McLaughlin, G. A., Jungreis, I., Fritch, E. J., Hou, Y. J., Ekena, J., Weidmann, C. A., Theesfeld, C. L., Kellis, M., Troyanskaya, O. G., Baric, R. S., Sheahan, T. P., Weeks, K. M. & Gladfelter, A. S. Genomic RNA Elements Drive Phase Separation of the SARS-CoV-2 Nucleocapsid. *Mol. Cell* **80**, 1078–1091.e6 (2020).
 12. Jack, A., Ferro, L. S., Trnka, M. J., Wehri, E., Nadgir, A., Nguyenla, X., Fox, D., Costa, K., Stanley, S., Schaletzky, J. & Yildiz, A. SARS-CoV-2 nucleocapsid protein forms condensates with viral genomic RNA. *PLOS Biol.* **19**, e3001425 (2021).
 13. Wootton, J. C. & Federhen, S. Analysis of compositionally biased regions in sequence databases. *Methods Enzymol.* **266**, 554–571 (1996).
 14. Rosenberg, G. M., Murray, K. A., Salwinski, L., Hughes, M. P., Abskharon, R. & Eisenberg, D. S. Bioinformatic identification of previously unrecognized amyloidogenic proteins. *J. Biol. Chem.* **298**, (2022).
 15. Martin, E. W. & Mittag, T. Relationship of Sequence and Phase Separation in Protein Low-Complexity Regions. *Biochemistry* **57**, 2478–2487 (2018).
 16. Chen, H., Cui, Y., Han, X., Hu, W., Sun, M., Zhang, Y., Wang, P. H., Song, G., Chen, W. & Lou, J. Liquid–liquid phase separation by SARS-CoV-2 nucleocapsid protein and RNA. *Cell Res.* **30**, 1143–1145 (2020).
 17. Carlson, C. R., Asfaha, J. B., Ghent, C. M., Howard, C. J., Hartooni, N., Safari, M., Frankel, A. D. & Morgan, D. O. Phosphoregulation of Phase Separation by the SARS-

- CoV-2 N Protein Suggests a Biophysical Basis for its Dual Functions. *Mol. Cell* **80**, 1092–1103.e4 (2020).
18. Bai, Z., Cao, Y., Liu, W. & Li, J. The SARS-CoV-2 Nucleocapsid Protein and Its Role in Viral Structure, Biological Functions, and a Potential Target for Drug or Vaccine Mitigation. *Viruses* **13**, 1115 (2021).
 19. Mullen L. Julia, Tsueng Ginger, Abdel Latif Alaa, Alkuzweny Manar, Cano Marco, Haag Emily, Zhou Jerry, Zeller Mark, Hufbauer Emory, Matteson Nate, Andersen G. Kristian, Wu Chunlei, Su I. Andrew, Gangavarapu Karthik, Hughes D. Laura & and the Center for Viral Systems Biology outbreak.info. outbreak.info. <https://outbreak.info/> (2020).
 20. Shu, Y. & McCauley, J. GISAID: Global initiative on sharing all influenza data – from vision to reality. *Eurosurveillance* **22**, 30494 (2017).
 21. Elbe, S. & Buckland-Merrett, G. Data, disease and diplomacy: GISAID’s innovative contribution to global health. *Glob. Challenges* **1**, 33–46 (2017).
 22. Syed, A. M., Taha, T. Y., Tabata, T., Chen, I. P., Ciling, A., Khalid, M. M., Sreekumar, B., Chen, P. Y., Hayashi, J. M., Soczek, K. M., Ott, M. & Doudna, J. A. Rapid assessment of SARS-CoV-2–evolved variants using virus-like particles. *Science* **374**, 1626–1632 (2021).
 23. Ong, S. W. X., Chiew, C. J., Ang, L. W., Mak, T.-M., Cui, L., Toh, M. P. H., Lim, Y. D., Lee, P. H., Lee, T. H., Chia, P. Y., Maurer-Stroh, S., Lin, R. T. P., Leo, Y.-S., Lee, V. J., Lye, D. C. & Young, B. E. Clinical and Virological Features of SARS-CoV-2 Variants of Concern: A Retrospective Cohort Study Comparing B.1.1.7 (Alpha), B.1.315 (Beta), and B.1.617.2 (Delta). *SSRN Electron. J.* doi:10.2139/ssrn.3861566 (2021).
 24. Sheikh, A., McMenamin, J., Taylor, B. & Robertson, C. SARS-CoV-2 Delta VOC in Scotland: demographics, risk of hospital admission, and vaccine effectiveness. *Lancet* **397**, 2461–2462 (2021).
 25. Dagpunar, J. Interim estimates of increased transmissibility, growth rate, and reproduction number of the Covid-19 B.1.617.2 variant of concern in the United Kingdom. *medRxiv* 2021.06.03.21258293 doi:10.1101/2021.06.03.21258293 (2021).
 26. Fisman, D. N. & Tuite, A. R. Evaluation of the relative virulence of novel SARS-CoV-2 variants: a retrospective cohort study in Ontario, Canada. *CMAJ* **193**, E1619–E1625 (2021).
 27. Furukawa, Y. & Nukina, N. Functional diversity of protein fibrillar aggregates from physiology to RNA granules to neurodegenerative diseases. *Biochim. Biophys. Acta - Mol. Basis Dis.* **1832**, 1271–1278 (2013).
 28. Ramaswami, M., Taylor, J. P. & Parker, R. Altered ribostasis: RNA-protein granules in degenerative disorders. *Cell* **154**, (2013).
 29. Molliex, A., Temirov, J., Lee, J., Coughlin, M., Kanagaraj, A. P., Kim, H. J., Mittag, T. & Taylor, J. P. Phase Separation by Low Complexity Domains Promotes Stress Granule Assembly and Drives Pathological Fibrillization. *Cell* **163**, 123–133 (2015).
 30. Goedert, M., Eisenberg, D. S. & Crowther, R. A. Propagation of Tau Aggregates and Neurodegeneration. *Annu. Rev. Neurosci.* **40**, 189–210 (2017).
 31. Eisenberg, D. & Jucker, M. The amyloid state of proteins in human diseases. *Cell* **148**, 1188–1203 (2012).
 32. Haikal, C., Pascual, L. O., Najarzadeh, Z., Bernfur, K., Svanbergsson, A., Otzen, D. E., Linse, S. & Li, J. Y. The bacterial amyloids phenol soluble modulins from staphylococcus aureus catalyze alpha-synuclein aggregation. *Int. J. Mol. Sci.* **22**, 11594 (2021).

33. Javed, I., Zhang, Z., Adamcik, J., Andrikopoulos, N., Li, Y., Otzen, D. E., Lin, S., Mezzenga, R., Davis, T. P., Ding, F. & Ke, P. C. Accelerated Amyloid Beta Pathogenesis by Bacterial Amyloid FapC. *Adv. Sci.* **7**, 2001299 (2020).
34. Sampson, T. R., Challis, C., Jain, N., Moiseyenko, A., Ladinsky, M. S., Shastri, G. G., Thron, T., Needham, B. D., Horvath, I., Debelius, J. W., Janssen, S., Knight, R., Wittung-Stafshede, P., Gradinaru, V., Chapman, M. & Mazmanian, S. K. A gut bacterial amyloid promotes α -synuclein aggregation and motor impairment in mice. *Elife* **9**, (2020).
35. Friedland, R. P. & Chapman, M. R. The role of microbial amyloid in neurodegeneration. *PLOS Pathog.* **13**, e1006654 (2017).
36. Ren, B., Zhang, Y., Zhang, M., Liu, Y., Zhang, D., Gong, X., Feng, Z., Tang, J., Chang, Y. & Zheng, J. Fundamentals of cross-seeding of amyloid proteins: An introduction. *J. Mater. Chem. B* **7**, 7267–7282 (2019).
37. Harrison, A. F. & Shorter, J. RNA-binding proteins with prion-like domains in health and disease. *Biochemical Journal* vol. 474 1417–1438 at <https://doi.org/10.1042/BCJ20160499> (2017).

REVIEWERS' COMMENTS

Reviewer #1 (Remarks to the Author):

I thank the authors for their revised article, the authors have addressed the vast majority of the points I raised in my original review, and have re-submitted a convincing and important piece of research. I agree with the authors decision to move the data on the self-assembly of the full protein to the SI and focus on the amyloidogenicity of the LCD, and their choice to remove the data claiming that G12 reduces NCAPs fibrillation propensity.

I would now recommend that this paper is published and my only remaining minor concern is with the authors response to the following:

"why do none of the proteins or fragments show +ve ThT signals in the absence of vRNA, the EM images of LCD -vRNA seem to show a high abundance of fibrils, you would expect this to result in some ThT signal?"

I accept the authors explanation that to detect fibrils by EM of the LCD in absence of RNA they needed to incubate the protein or peptides at higher concentrations for longer periods of time. In addition to this it would have been preferable to perform a quick ThT end point measurement to show that these self-assembled LCD structures are ThT positive. This would have proven beyond reasonable doubt that the assembled LCD structures both with and without RNA are amyloid in nature.

Reviewer #2 (Remarks to the Author):

The authors have addressed all of my concerns, and I support publication.

Reviewer #3 (Remarks to the Author):

The revised version of this manuscript by Tayeb-Fligelman et al. is improved. It's appropriate that the authors removed the claims and statements from the result section which were not supported by their previous experimental data in the revision. I don't have further question.

RE: Low Complexity Domains of the Nucleocapsid Protein of SARS-CoV-2 Form Amyloid Fibrils

We thank the reviewers for their positive feedback. Below we address the point raised by reviewer #1.

Reviewer #1 (Remarks to the Author):

I thank the authors for their revised article, the authors have addressed the vast majority of the points I raised in my original review, and have re-submitted a convincing and important piece of research. I agree with the authors decision to move the data on the self-assembly of the full protein to the SI and focus on the amyloidogenicity of the LCD, and their choice to remove the data claiming that G12 reduces NCAPs fibrillation propensity. I would now recommend that this paper is published and my only remaining minor concern is with the authors response to the following:

"why do none of the proteins or fragments show +ve ThT signals in the absence of vRNA, the EM images of LCD -vRNA seem to show a high abundance of fibrils, you would expect this to result in some ThT signal?"

I accept the authors explanation that to detect fibrils by EM of the LCD in absence of RNA they needed to incubate the protein or peptides at higher concentrations for longer periods of time. In addition to this it would have been preferable to perform a quick ThT end point measurement to show that these self-assembled LCD structures are ThT positive. This would have proven beyond reasonable doubt that the assembled LCD structures both with and without RNA are amyloid in nature.

We thank the reviewer for suggesting this experiment. We measured the endpoint ThT signal of three concentrated LCD only samples versus buffer only controls at days 1, 6 and 11 of incubation. The ThT signal of LCD only samples increase over time, unlike the buffer controls, but with large sample to sample variability. We now present those results in Figure 1. g. We also accordingly revised the main text in lines 123- 129 in the final version of the paper and added the appropriate method paragraph.

Overall, ThT binding to the LCD segment in the absence of RNA was slower and lower than in its presence. Nevertheless, the amyloid-typical fiber diffraction that we show for the LCD only fibrils (Figure 2. a) is a further support for the amyloid nature of those fibrils. We therefore hope

to convince the reviewer and readers that the LCD segment of NCAP forms amyloid fibrils in the absence and presence of RNA.

Lines 123- 129 now read:

“..with increased protein concentration and incubation time, fibrils were detected by EM both in the presence and absence of the vRNA. Fibrils of the DD-Cterm segment with vRNA are morphologically different than those grown in its absence, however, the central LCD segment produces amyloid-looking fibrils under both conditions. Indeed, concentrated LCD only samples exhibit increased ThT fluorescence signal upon 6 and 11 days of incubation, but with large sample to sample variability (Figure 1. g). vRNA is, therefore, not essential for fibril formation and ThT binding, but may promote these processes.”

The results of our endpoint ThT experiment (now panel g of figure 1):

Legend of figure 1.g: “Endpoint ThT fluorescence measurements of concentrated LCD only samples (pink) and buffer only controls (white) at days 1, 6, and 11 of incubation. Dots indicate individual data points and bars present mean values +/- SD. n= 3 samples.”

Reviewer #2 (Remarks to the Author):

The authors have addressed all of my concerns, and I support publication.

We thank the reviewer for supporting the publication of our manuscript.

Reviewer #3 (Remarks to the Author):

The revised version of this manuscript by Tayeb-Fligelman et al. is improved. It's appropriate that the authors removed the claims and statements from the result section which were not supported by their previous experimental data in the revision. I don't have further question.

We thank the reviewer for the positive feedback.